# Measuring cis-regulatory energetics in living cells using allelic manifolds

**Talitha L Forcier[1], Andalus Ayaz[1], Manraj S Gill[1†], Daniel Jones[1,2‡], Rob Phillips[2], Justin B Kinney[1]\***

[1]Simons Center for Quantitative Biology, Cold Spring Harbor Laboratory, Cold Spring Harbor, United States; [2]Department of Applied Physics, California Institute of Technology, Pasadena, United States

**Abstract** Gene expression in all organisms is controlled by cooperative interactions between DNA-bound transcription factors (TFs), but quantitatively measuring TF-DNA and TF-TF interactions remains difficult. Here we introduce a strategy for precisely measuring the Gibbs free energy of such interactions in living cells. This strategy centers on the measurement and modeling of 'allelic manifolds', a multidimensional generalization of the classical genetics concept of allelic series. Allelic manifolds are measured using reporter assays performed on strategically designed cis-regulatory sequences. Quantitative biophysical models are then fit to the resulting data. We used this strategy to study regulation by two *Escherichia coli* TFs, CRP and $\sigma^{70}$ RNA polymerase. Doing so, we consistently obtained energetic measurements precise to $\sim 0.1$ kcal/mol. We also obtained multiple results that deviate from the prior literature. Our strategy is compatible with massively parallel reporter assays in both prokaryotes and eukaryotes, and should therefore be highly scalable and broadly applicable.

**Editorial note:** This article has been through an editorial process in which the authors decide how to respond to the issues raised during peer review. The Reviewing Editor's assessment is that minor issues remain unresolved (see decision letter).

DOI: https://doi.org/10.7554/eLife.40618.001

**\*For correspondence:**
jkinney@cshl.edu

**Present address:** †Department of Biology, Massachusetts Institute of Technology, Massachusetts, United States; ‡Department of Cell and Molecular Biology, Uppsala University, Uppsala, Sweden

**Competing interests:** The authors declare that no competing interests exist.

## Introduction

Cells regulate the expression of their genes in response to biological and environmental cues. A major mechanism of gene regulation in all organisms is the binding of transcription factor (TF) proteins to cis-regulatory elements encoded within genomic DNA. DNA-bound TFs interact with one another, either directly or indirectly, forming cis-regulatory complexes that modulate the rate at which nearby genes are transcribed (*Ptashne and Gann, 2002*; *Courey, 2008*). Different arrangements of TF binding sites within cis-regulatory sequences can lead to different regulatory programs, but the rules that govern *which* arrangements lead to *which* regulatory programs remain largely unknown. Understanding these rules, which are often referred to as 'cis-regulatory grammar' (*Spitz and Furlong, 2012*), is a major challenge in modern biology.

Measuring the quantitative strength of interactions among DNA-bound TFs is critical for elucidating cis-regulatory grammar. In particular, knowing the Gibbs free energy of TF-DNA and TF-TF interactions is essential for building biophysical models that can quantitatively explain gene regulation in terms of simple protein-DNA and protein-protein interactions (*Shea and Ackers, 1985*; *Bintu et al., 2005*; *Sherman and Cohen, 2012*). Biophysical models have proven remarkably successful at quantitatively explaining regulation by a small number of well-studied cis-regulatory sequences. Arguably, the biggest successes have been achieved in the bacterium *Escherichia coli*, particularly in the context of the *lac* promoter (*Vilar and Leibler, 2003*; *Kuhlman et al., 2007*; *Kinney et al., 2010*; *Garcia and Phillips, 2011*; *Brewster et al., 2014*) and the $O_R/O_L$ control region of the $\lambda$ phage

lysogen (*Ackers et al., 1982*; *Shea and Ackers, 1985*; *Cui et al., 2013*). But in both cases, this quantitative understanding has required decades of focused study. New approaches for dissecting cis-regulatory energetics, approaches that are both systematic and scalable, will be needed before a general quantitative understanding of cis-regulatory grammar can be developed.

Here we address this need by describing a systematic experimental/modeling strategy for dissecting the biophysical mechanisms of transcriptional regulation in living cells. Our strategy centers on the concept of an 'allelic manifold'. Allelic manifolds generalize the classical genetics concept of allelic series to multiple dimensions. An allelic series is a set of sequence variants that affect the same phenotype (or phenotypes) but differ in their quantitative strength. Here we construct allelic manifolds by measuring, in *multiple* experimental contexts, the phenotypic strength of each variant in an allelic series. Each variant thus corresponds to a data point in a multi-dimensional 'measurement space'. If the measurement space is of high enough dimension, and if one's measurements are sufficiently precise, these data should collapse to a lower-dimension manifold that represents the inherent phenotypic dimensionality of the allelic series. These data can then be used to infer quantitative biophysical models that describe the shape of the allelic manifold, as well as the location of each allelic variant within that manifold. As we show here, such inference allows one to determine in vivo values for important biophysical quantities with remarkable precision.

We demonstrate this strategy on a regulatory paradigm in *E. coli*: activation of the $\sigma^{70}$ RNA polymerase holoenzyme (RNAP) by the cAMP receptor protein (CRP, also called CAP). CRP activates transcription when bound to DNA at positions upstream of RNAP (*Busby and Ebright, 1999*), and the strength of these interactions is known to depend strongly on the precise nucleotide spacing between CRP and RNAP binding sites (*Gaston et al., 1990*; *Ushida and Aiba, 1990*). However, the Gibbs free energies of these interactions are still largely unknown. To our knowledge, only the CRP-RNAP interaction at the *lac* promoter has previously been quantitatively measured (*Kuhlman et al., 2007*; *Kinney et al., 2010*). By measuring and modeling allelic manifolds, we systematically determined the in vivo Gibbs free energy ($\Delta G$) of CRP-RNAP interactions that occur at a variety of different binding site spacings. These $\Delta G$ values were consistently measured to an estimated precision of $\sim 0.1$ kcal/mol. We also obtained $\Delta G$ values for in vivo CRP-DNA and RNAP-DNA interactions, again with similar estimated precision.

The Results section that follows is organized into three Parts, each of which describes a different use for allelic manifolds. Part 1 focuses on measuring TF-DNA interactions, Part 2 focuses on TF-TF interactions, and Part 3 shows how to distinguish different possible mechanisms of transcriptional activation. Each Part consists of three subsections: Strategy, Demonstration, and Aside. Strategy covers the theoretical basis for the proposed use of allelic manifolds. Demonstration describes how we applied this strategy to better understand regulation by CRP and RNAP. Aside describes related findings that are interesting but somewhat tangential.

## Results

### Part 1. Strategy: Measuring TF-DNA interactions

We begin by showing how allelic manifolds can be used to measure the in vivo strength of TF binding to a specific DNA binding site. This measurement is accomplished by using the TF of interest as a transcriptional repressor. We place the TF binding site directly downstream of the RNAP binding site in a bacterial promoter so that the TF, when bound to DNA, sterically occludes the binding of RNAP. We then measure the rate of transcription from a few dozen variant RNAP binding sites. Transcription from each variant site is assayed in both the presence and in the absence of the TF.

*Figure 1A* illustrates a thermodynamic model (*Shea and Ackers, 1985*; *Bintu et al., 2005*; *Sherman and Cohen, 2012*) for this type of simple repression. In this model, promoter DNA can be in one of three states: unbound, bound by the TF, or bound by RNAP. Each of these three states is assumed to occur with a frequency that is consistent with thermal equilibrium, that is with a probability proportional to its Boltzmann weight.

The energetics of protein-DNA binding determine the Boltzmann weight for each state. By convention we set the weight of the unbound state equal to 1. The weight of the TF-bound state is then given by $F = [\text{TF}]K_F$ where $[\text{TF}]$ is the concentration of the TF and $K_F$ is the affinity constant in inverse molar units. Similarly, the weight of the RNAP-bound state is $P = [\text{RNAP}]K_P$. In what follows

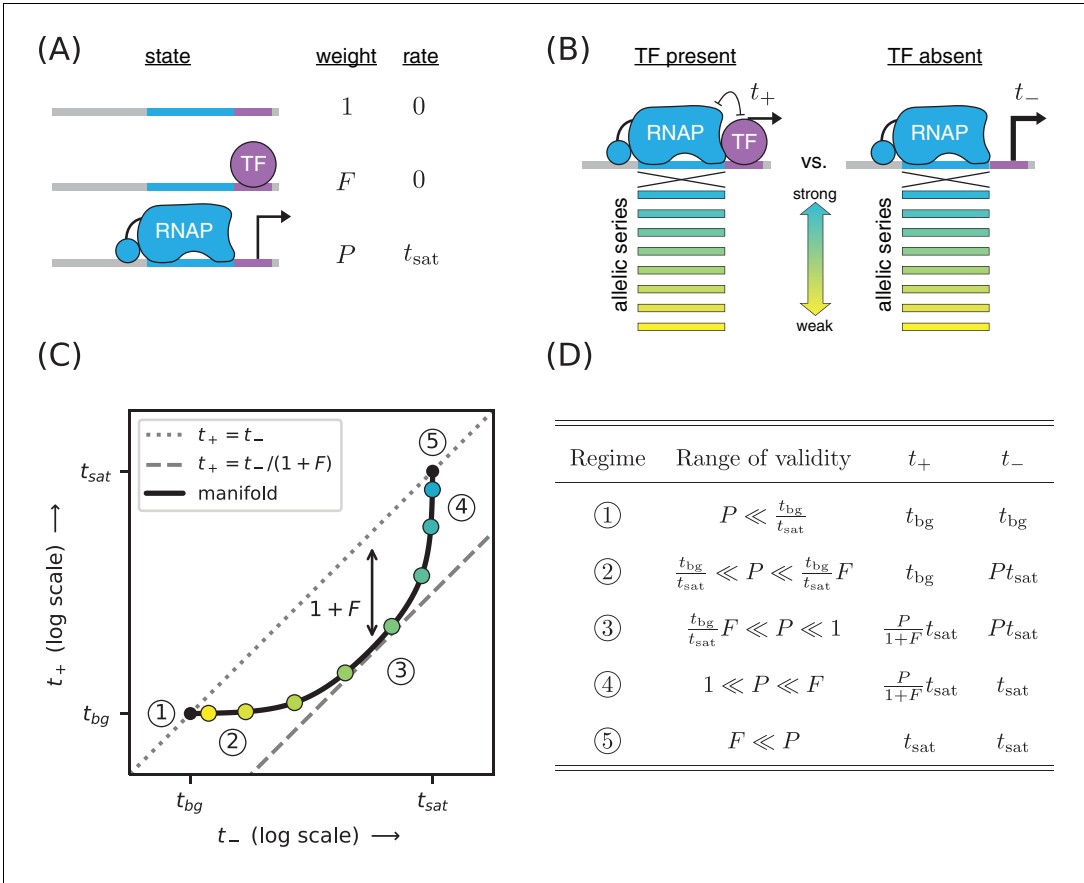

**Figure 1.** Strategy for measuring TF-DNA interactions. (**A**) A thermodynamic model of simple repression. Here, promoter DNA can transition between three possible states: unbound, bound by a TF, or bound by RNAP. Each state has an associated Boltzmann weight and rate of transcript initiation. $F$ is the TF binding factor and $P$ is the RNAP binding factor; see text for a description of how these dimensionless binding factors relate to binding affinity and binding energy. $t_{\text{sat}}$ is the rate of specific transcript initiation from a promoter fully occupied by RNAP. (**B**) Transcription is measured in the presence ($t_+$) and absence ($t_-$) of the TF. Measurements are made for an allelic series of RNAP binding sites that differ in their binding strengths (blue-yellow gradient). (**C**) If the model in panel A is correct, plotting $t_+$ vs. $t_-$ for the promoters in panel B (colored dots) will trace out a 1D allelic manifold. Mathematically, this manifold reflects *Equation 1* and *Equation 2* computed over all possible values of the RNAP binding factor $P$ while the other parameters ($F$, $t_{\text{sat}}$) are held fixed. Note that these equations include a background transcription term $t_{\text{bg}}$; it is assumed throughout that $t_{\text{bg}} \ll t_{\text{sat}}$ and that $t_{\text{bg}}$ is independent of RNAP binding site sequence. The resulting manifold exhibits five distinct regimes (circled numbers), corresponding to different ranges for the value of $P$ that allow the mathematical expressions in *Equations 1 and 2* to be approximated by simplified expressions. In regime 3, for instance, $t_+ \approx t_-/(1+F)$, and thus the manifold approximately follows a line parallel (on a log-log plot) to the diagonal but offset below it by a factor of $1+F$ (dashed line). Data points in this regime can therefore be used to determine the value of $F$. (**D**) The five regimes of the allelic manifold, including approximate expressions for $t_+$ and $t_-$ in each regime, as well as the range of validity for $P$.

DOI: https://doi.org/10.7554/eLife.40618.002

we refer to $F$ and $P$ as the 'binding factors' of the TF-DNA and RNAP-DNA interactions, respectively. We note that these binding factors can also be written as $F = e^{-\Delta G_F/k_B T}$ and $P = e^{-\Delta G_P/k_B T}$ where $k_B$ is Boltzmann's constant, $T$ is temperature, and $\Delta G_F$ and $\Delta G_P$ respectively denote the Gibbs free energy of binding for the TF and RNAP. Note that each Gibbs free energy accounts for the entropic cost of pulling each protein out of solution. In what follows, we report $\Delta G$ values in units of kcal/mol; note that 1 kcal/mol = $1.62\ k_B T$ at 37 °C.

The overall rate of transcription is computed by summing the amount of transcription produced by each state, weighting each state by the probability with which it occurs. In this case we assume the RNAP-bound state initiates at a rate of $t_{\text{sat}}$, and that the other states produce no transcripts. We also add a term, $t_{\text{bg}}$, to account for background transcription (e.g., from an unidentified promoter further upstream). The rate of transcription in the presence of the TF is thus given by

$$t_+ = t_{\text{sat}} \frac{P}{1 + F + P} + t_{\text{bg}}. \tag{1}$$

In the absence of the TF ($F = 0$), the rate of transcription becomes

$$t_- = t_{\text{sat}} \frac{P}{1 + P} + t_{\text{bg}}. \tag{2}$$

Our goal is to measure the TF-DNA binding factor $F$. To do this, we create a set of promoter sequences where the RNAP binding site is varied (thus generating an allelic series) but the TF binding site is kept fixed. We then measure transcription from these promoters in both the presence and absence of the TF, respectively denoting the resulting quantities by $t_+$ and $t_-$ (*Figure 1B*). Our rationale for doing this is that changing the RNAP binding site sequence should, according to our model, affect only the RNAP-DNA binding factor $P$. All of our measurements are therefore expected to lie along a one-dimensional allelic manifold residing within the two-dimensional space of $(t_-, t_+)$ values. Moreover, this allelic manifold should follow the specific mathematical form implied by *Equations 1 and 2* when $P$ is varied and the other parameters ($t_{\text{sat}}, t_{\text{bg}}, F$) are held fixed; see *Figure 1C*.

The geometry of this allelic manifold is nontrivial. Assuming $F \gg 1$ and $t_{\text{bg}} \ll t_{\text{sat}}$, there are five different regimes corresponding to different values of the RNAP binding factor $P$. These regimes are listed in *Figure 1D* and derived in Appendix 4. In regime 1, $P$ is so small that both $t_+$ and $t_-$ are dominated by background transcription, that is $t_+ \approx t_- \approx t_{\text{bg}}$. $P$ is somewhat larger in regime 2, causing $t_-$ to be proportional to $P$ while $t_+$ remains dominated by background. In regime 3, both $t_+$ and $t_-$ are proportional to $P$ with $t_+/t_- \approx 1/(1 + F)$. In regime 4, $t_-$ saturates at $t_{\text{sat}}$ while $t_+$ remains proportional to $P$. Regime five occurs when both $t_+$ and $t_-$ are saturated, that is $t_+ \approx t_- \approx t_{\text{sat}}$.

## Part 1. Demonstration: Measuring CRP-DNA binding

The placement of CRP immediately downstream of RNAP is known to repress transcription (*Morita et al., 1988*). We therefore reasoned that placing a DNA binding site for CRP downstream of RNAP would allow us to measure the binding factor of that site. *Figure 2* illustrates measurements of the allelic manifold used to characterize the strength of CRP binding to the 22 bp site GAA<u>TGTGA</u>CCTAGA<u>TCACA</u>TTT. This site contains the well-known consensus site, which comprises two palindromic pentamers (underlined) separated by a 6 bp spacer (*Gunasekera et al., 1992*). We performed measurements using this CRP site centered at two different locations relative to the transcription start site (TSS): +0.5 bp and +4.5 bp. Note that the first transcribed base is, in this paper, assigned position 0 instead of the more conventional +1, and half-integer positions indicate centering between neighboring nucleotides. To avoid influencing CRP binding strength, the −10 region of the RNAP site was kept fixed in the promoters we assayed while the −35 region of the RNAP binding site was varied (*Figure 2A*). Promoter DNA sequences are shown in *Appendix 1—figure 1*.

We obtained $t_-$ and $t_+$ measurements for these constructs using a modified version of the colorimetric β-galactosidase assay of *Lederberg (1950)* and *Miller (1972)*; see Appendix 2 for details. Our measurements are largely consistent with an allelic manifold having the expected mathematical form (*Figure 2B*). Moreover, the measurements for promoters with CRP sites at two different positions (+0.5 bp and +4.5 bp) appear consistent with each other, although the measurements for +4.5 bp promoters appear to have lower values for $P$ overall. A small number of data points do deviate substantially from this manifold, but the presence of such outliers is not surprising from a biological perspective (see Discussion). Fortunately, outliers appear at a rate small enough for us to identify them by inspection.

We quantitatively modeled the allelic manifold in *Figure 2B* by fitting $n + 3$ parameters to our $2n$ measurements, where $n = 39$ is the number of non-outlier promoters. The $n + 3$ parameters were $t_{\text{sat}}$, $t_{\text{bg}}$, $F$, and $P_1, P_2, \ldots, P_n$, where each $P_i$ is the RNAP binding factor of promoter $i$. Nonlinear least squares optimization was used to infer values for these parameters. Uncertainties in $t_{\text{sat}}$, $t_{\text{bg}}$, and $F$

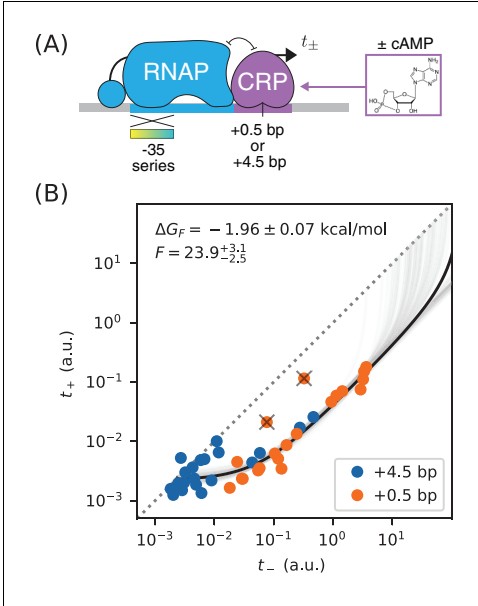

**Figure 2.** Precision measurement of in vivo CRP-DNA binding. (**A**) Expression measurements were performed on promoters for which CRP represses transcription by occluding RNAP. Each promoter assayed contained a near-consensus CRP binding site centered at either +0.5 bp or +4.5 bp, as well as an RNAP binding site with a partially mutagenized −35 region (gradient). $t_+$ (or $t_-$) denotes measurements made using *E. coli* strain JK10 grown in the presence (or absence) of the small molecule cAMP. (**B**) Dots indicate measurements for 41 such promoters. A best-fit allelic manifold (black) was inferred from $n = 39$ of these data points after the exclusion of 2 outliers (gray 'X's). Gray lines indicate 100 plausible allelic manifolds fit to bootstrap-resampled data points. The parameters of these manifolds were used to determine the CRP-DNA binding factor $F$ and thus the Gibbs free energy $\Delta G_F = -k_B T \log F$. Error bars indicate 68% confidence intervals determined by bootstrap resampling. See Appendix 3 for more information about our manifold fitting procedure.

DOI: https://doi.org/10.7554/eLife.40618.003

were quantified by repeating this procedure on bootstrap-resampled data points. See Appendix 3 for details.

These results yielded highly uncertain values for $t_{\text{sat}}$ because none of our measurements appear to fall within regime 4 or 5 of the allelic manifold. A reasonably precise value for $t_{\text{bg}}$ was obtained, but substantial scatter about our model predictions in regime 1 and 2 remain. This scatter likely reflects some variation in $t_{\text{bg}}$ from promoter to promoter, variation that is to be expected since the source of background transcription is not known and the appearance of even very weak promoters could lead to such fluctuations.

These data do, however, determine a highly precise value for the strength of CRP-DNA binding: $F = 23.9^{+3.1}_{-2.5}$ or, equivalently, $\Delta G_F = -1.96 \pm 0.07$ kcal/mol. This allelic manifold approach is thus able to measure the strength of TF-DNA binding with a precision of $\sim 0.1$ kcal/mol. For comparison, the typical strength of a hydrogen bond in liquid water is $-1.9$ kcal/mol (*Markovitch and Agmon, 2007*).

We note that CRP forms approximately 38 hydrogen bonds with DNA when it binds to a consensus DNA site (*Parkinson et al., 1996*). Our result indicates that, in living cells, the enthalpy resulting from these and other interactions is almost exactly canceled by entropic factors. We also note that our in vivo value for $F$ is far smaller than expected from experiments in aqueous solution. The consensus CRP binding site has been measured in vitro to have an affinity constant of $K_F \sim 10^{11}$ M$^{-1}$ (*Ebright et al., 1989*). There are probably about $10^3$ CRP dimers per cell (*Schmidt et al., 2016*), giving a concentration $[\text{CRP}] \sim 10^{-6}$ M. Putting these numbers together gives a binding factor of $F \sim 10^5$. The nonspecific binding of CRP to genomic DNA and other molecules in the cell, and perhaps limited DNA accessibility as well, might be responsible for this $\sim 10^5$-fold disagreement with our in vivo measurements.

## Part 1. Aside: Measuring changes in the concentration of active CRP

Varying cAMP concentrations in growth media changes the in vivo concentration of active CRP in the *E. coli* strain we assayed (JK10). Such variation is therefore expected to alter the CRP-DNA binding factor $F$. We tested whether this was indeed the case by measuring multiple allelic manifolds, each using a different concentration of [cAMP] when measuring $t_+$. These measurements were performed on promoters with CRP binding sites at +0.5 bp (*Figure 3A*). The resulting data are shown in *Figure 3B*. To these data, we fit allelic manifolds having variable values for $F$, but fixed values for both $t_{\text{bg}}$ and $t_{\text{sat}}$ ($t_{\text{bg}} = 2.30 \times 10^{-3}$ a.u. was inferred in the prior analysis for *Figure 2B*; $t_{\text{sat}} = 15.1$ a.u. was inferred in the subsequent analysis for Figure 5C).

This procedure allowed us to quantitatively measure changes in the RNAP binding factor $F$, and thus changes in the in vivo concentration of active CRP. Our results, shown in *Figure 3C*, suggest a

nontrivial power law relationship between $F$ and [cAMP]. To quantify this relationship, we performed least squares regression ($\log F$ against $\log$ [cAMP]) using data for the four largest cAMP concentrations; measurements of $F$ for the three other cAMP concentrations have large asymmetric uncertainties and were therefore excluded. We found that $F \propto [\text{cAMP}]^{1.41 \pm 0.18}$, with error bars representing a 95% confidence interval. We emphasize, however that our data do not rule out a more complex relationship between [cAMP] and $F$.

There are multiple potential explanations for this deviation from proportionality. One possibility is cooperative binding of cAMP to the two binding sites within each CRP dimer. Such cooperativity could, for instance, result from allosteric effects like those described in *Einav et al., 2018*. Alternatively, this power law behavior might reflect unknown aspects of how cAMP is imported and exported from *E. coli* cells. It is worth comparing and contrasting this result to those reported in *Kuhlman et al. (2007)*. JK10, the *E. coli* strain used in our experiments, is derived from strain TK310, which was developed in *Kuhlman et al. (2007)*. In that work, the authors concluded that $F \propto [\text{cAMP}]$, whereas our data leads us to reject this hypothesis. This illustrates one way in which using allelic manifolds to measure how in vivo TF concentrations vary with growth conditions can be useful.

## Part 2. Strategy: Measuring TF-RNAP interactions

Next we discuss how to measure an activating interaction between a DNA-bound TF and DNA-bound RNAP. A common mechanism of transcriptional activation is 'stabilization' (also called 'recruitment'; see *Ptashne, 2003*). This occurs when a DNA-bound TF stabilizes the RNAP-DNA closed complex. Stabilization effectively increases the RNAP-DNA binding affinity $K_P$, and thus the binding factor $P$. It does not affect $t_{\text{sat}}$, the rate of transcript initiation from RNAP-DNA closed complexes.

A thermodynamic model for activation by stabilization is illustrated in *Figure 4A*. Here promoter DNA can be in four states: unbound, TF-bound, RNAP-bound, or doubly bound. In the doubly bound state, a 'cooperativity factor' $\alpha$ contributes to the Boltzmann weight. This cooperativity factor

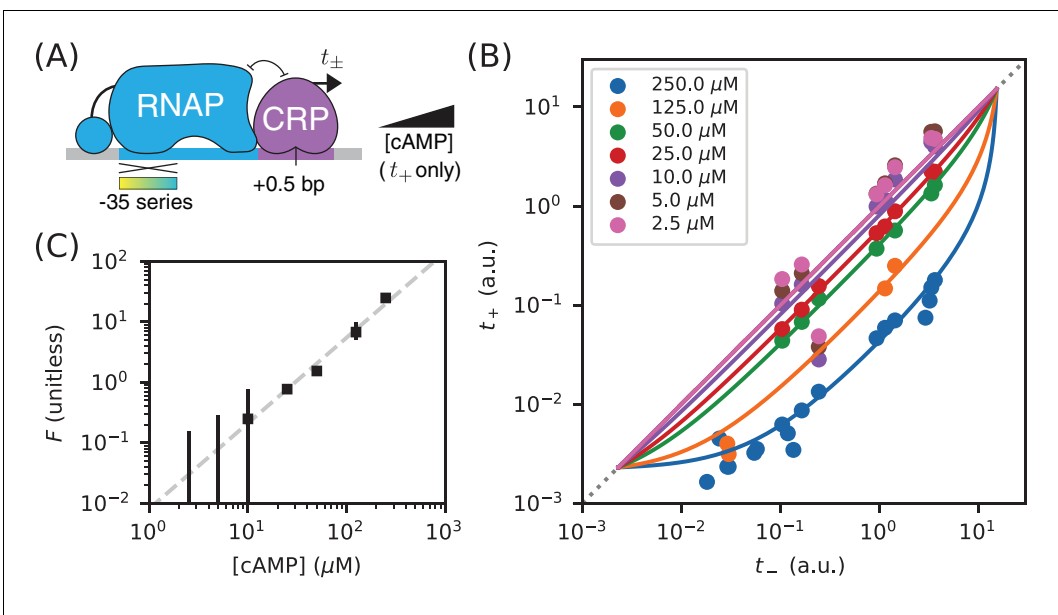

**Figure 3.** Measuring in vivo changes in TF concentration. (**A**) Allelic manifolds were measured for the +0.5 bp occlusion promoter architecture using seven different concentrations of cAMP (ranging from 2.5 μM to 250 μM) when assaying $t_+$. (**B**) As expected, these data follow allelic manifolds that have cAMP-dependent values for the CRP binding factor $F$. (**C**) Values for $F$ inferred from the data in panel B exhibit a nontrivial power law dependence on [cAMP]. Error bars indicate 68% confidence intervals determined by bootstrap resampling.
DOI: https://doi.org/10.7554/eLife.40618.004

is related to the TF-RNAP Gibbs free energy of interaction, $\Delta G_\alpha$, via $\alpha = e^{-\Delta G_\alpha/k_B T}$. Activation occurs when $\alpha > 1$ (i.e., $\Delta G_\alpha < 0$). The resulting activated transcription rate is given by

$$t_+ = t_{\text{sat}} \frac{P + \alpha FP}{1 + F + P + \alpha FP} + t_{\text{bg}}. \tag{3}$$

This can be rewritten as

$$t_+ = t_{\text{sat}} \frac{\alpha' P}{1 + \alpha' P} + t_{\text{bg}}, \tag{4}$$

where

$$\alpha' = \frac{1 + \alpha F}{1 + F} \tag{5}$$

is a renormalized cooperativity that accounts for the strength of TF-DNA binding. As before, $t_-$ is given by *Equation 2*. Note that $\alpha' \leq \alpha$ and that $\alpha' \approx \alpha$ when $F \gg 1$ and $\alpha \gg 1/F$.

As before, we measure both $t_+$ and $t_-$ for an allelic series of RNAP binding sites (*Figure 4B*). These measurements will, according to our model, lie along an allelic manifold resembling the one shown in *Figure 4C*. This allelic manifold exhibits five distinct regimes (when $t_{\text{sat}}/t_{\text{bg}} \gg \alpha' \gg 1$), which are listed in *Figure 4D*.

## Part 2. Demonstration: Measuring class I CRP-RNAP interactions

CRP activates transcription at the *lac* promoter and at other promoters by binding to a 22 bp site centered at −61.5 bp relative to the TSS. This is an example of class I activation, which is mediated by an interaction between CRP and the C-terminal domain of one of the two RNAP $\alpha$ subunits (the $\alpha$ CTDs) (*Busby and Ebright, 1999*). In vitro experiments have shown this class I CRP-RNAP interaction to activate transcription by stabilizing the RNAP-DNA closed complex.

We measured $t_+$ and $t_-$ for 47 variants of the lac* promoter (see *Appendix 1—figure 1* for sequences). These promoters have the same CRP binding site assayed for *Figure 2*, but positioned at −61.5 bp relative to the TSS (*Figure 5A*). They differ from one another in the −10 or −35 regions of their RNAP binding sites. *Figure 5B* shows the resulting measurements. With the exception of 3 outlier points, these measurements appear consistent with stabilizing activation via a Gibbs free energy of $\Delta G_\alpha = -4.05 \pm 0.08$ kcal/mol, corresponding to a cooperativity of $\alpha = 712^{+102}_{-83}$. We note that, with $F = 23.9$ determined in *Figure 2B*, $\alpha' = \alpha$ to 4% accuracy.

This observed cooperativity is substantially stronger than suggested by previous work. Early in vivo experiments suggested a much lower cooperativity value, for example 50-fold (*Beckwith et al., 1972*), 20-fold (*Ushida and Aiba, 1990*), or even 10-fold (*Gaston et al., 1990*). These previous studies, however, only measured the ratio $t_+/t_-$ for a specific choice of RNAP binding site. This ratio is (by *Equation 4*) always less than $\alpha$ and the differences between these quantities can be substantial. However, even studies that have used explicit biophysical modeling have determined lower cooperativity values: *Kuhlman et al. (2007)* reported a cooperativity of $\alpha \approx 240$ ($\Delta G_\alpha \approx -3.4$ kcal/mol), while *Kinney et al. (2010)* reported $\alpha \approx 220$ ($\Delta G_\alpha \approx -3.3$ kcal/mol). Both of these studies, however, relied on the inference of complex biophysical models with many parameters. The allelic manifold in *Figure 4*, by contrast, is characterized by only three parameters ($t_{\text{sat}}$, $t_{\text{bg}}$, $\alpha'$), all of which can be approximately determined by visual inspection.

To test the generality of this approach, we measured allelic manifolds for 11 other potential class I promoter architectures. At every one of these positions we clearly observed the collapse of data to a 1D allelic manifold of the expected shape (*Figure 5C*). We then modeled these data using values of $\alpha$ and $t_{\text{bg}}$ that depend on CRP binding site location, as well as a single overall value for $t_{\text{sat}}$. The resulting values for $\alpha$ (and equivalently $\Delta G_\alpha$) are shown in *Figure 5D* and reported in Table 1. As first shown by *Gaston et al. (1990)* and *Ushida and Aiba (1990)*, $\alpha$ depends strongly on the spacing between the CRP and RNAP binding sites. In particular, $\alpha$ exhibits a strong ~ 10.5 bp periodicity reflecting the helical twist of DNA. However, as with the measurement in *Figure 5B*, the $\alpha$ values we measure are far larger than the $t_+/t_-$ ratios previously reported by *Gaston et al. (1990)* and *Ushida and Aiba (1990)*; see *Table 1*. We also find $t_{\text{sat}} = 15.1^{+0.6}_{-0.5}$ a.u. The single-cell observations of *So et al. (2011)* suggest that this corresponds to $13.8 \pm 6.6$ transcripts per minute. By pure

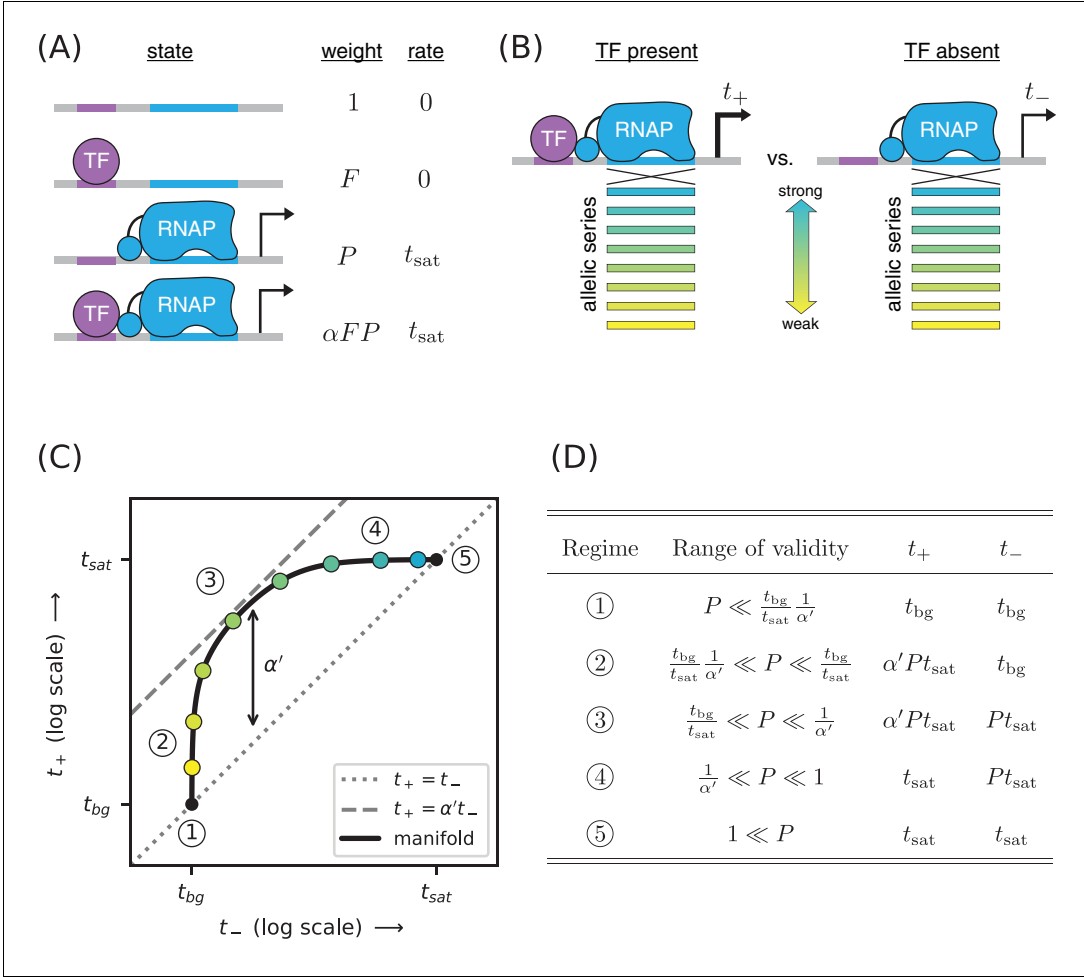

**Figure 4.** Strategy for measuring TF-RNAP interactions. (A) A thermodynamic model of simple activation. Here, promoter DNA can transition between four different states: unbound, bound by the TF, bound by RNAP, or doubly bound. As in *Figure 1*, $F$ is the TF binding factor, $P$ is the RNAP binding factor, and $t_{\text{sat}}$ is the rate of transcript initiation from an RNAP-saturated promoter. The cooperativity factor $\alpha$ quantifies the strength of the interaction between DNA-bound TF and RNAP molecules; see text for more information on this quantity. (B) As in *Figure 1*, expression is measured in the presence ($t_+$) and absence ($t_-$) of the TF for promoters that have an allelic series of RNAP binding sites (blue-yellow gradient). (C) If the model in panel A is correct, plotting $t_+$ vs. $t_-$ (colored dots) will reveal a 1D allelic manifold that corresponds to *Equation 4* (for $t_+$) and *Equation 2* (for $t_-$) evaluated over all possible values of $P$. Circled numbers indicate the five regimes of this manifold. In regime 3, $t_+ \approx \alpha' t_-$ where $\alpha'$ is the renormalized cooperativity factor given in *Equation 5*; data in this regime can thus be used to measure $\alpha'$. Separate measurements of $F$, using the strategy in *Figure 1*, then allow one to compute $\alpha$ from knowledge of $\alpha'$. (D) The five regimes of the allelic manifold in panel C. Note that these regimes differ from those in *Figure 1D*.

DOI: https://doi.org/10.7554/eLife.40618.005

coincidence, the 'arbitrary unit' (a.u.) units we use in this paper correspond very closely to 'transcripts per minute'.

## Part 2. Aside: Difficulties predicting binding affinity from DNA sequence

The measurement and modeling of allelic manifolds sidesteps the need to parametrically model how protein-DNA binding affinity depends on DNA sequence. In modeling the allelic manifolds in *Figure 5C*, we obtained values for the RNAP binding factor, $P = [\text{RNAP}]K_P$, for each variant RNAP binding site from the position of the corresponding data point along the length of the manifold.

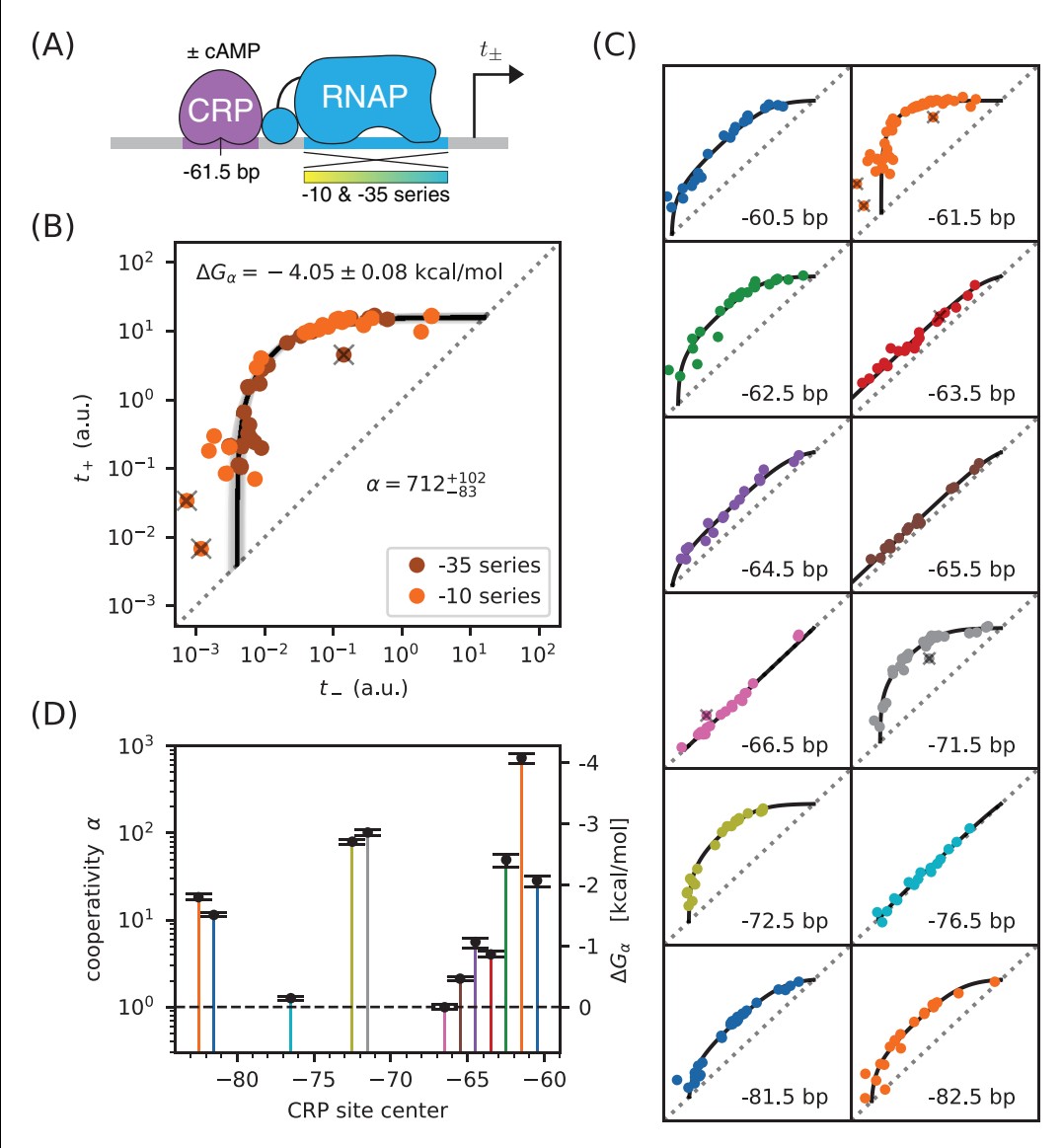

**Figure 5.** Precision measurement of class I CRP-RNAP interactions. (A) $t_+$ and $t_-$ were measured for promoters containing a CRP binding site centered at $-61.5$ bp. The RNAP sites of these promoters were mutagenized in either their $-10$ or $-35$ regions (gradient), generating two allelic series. As in *Figure 2*, $t_+$ and $t_-$ correspond to expression measurements respectively made in the presence and absence of cAMP. (B) Data obtained for 47 variant promoters having the architecture shown in panel A. Three data points designated as outliers are indicated by 'X's. The allelic manifold that best fits the $n = 44$ non-outlier points is shown in black; 100 plausible manifolds, estimated from bootstrap-resampled data points, are shown in gray. The resulting values for $\alpha$ and $\Delta G_\alpha = -k_B T \log \alpha$ are also shown, with 68% confidence intervals indicated. (C) Allelic manifolds obtained for promoters with CRP binding sites centered at a variety of class I positions. (D) Inferred values for the cooperativity factor $\alpha$ and corresponding Gibbs free energy $\Delta G_\alpha$ for the 12 different promoter architectures assayed in panel C. Error bars indicate 68% confidence intervals. Numerical values for $\alpha$ and $\Delta G_\alpha$ at all of these class I positions are provided in Table 1.

DOI: https://doi.org/10.7554/eLife.40618.006

RNAP has a very well established sequence motif (*McClure et al., 1983*). Indeed, its DNA binding requirements were among the first characterized for any DNA-binding protein (*Pribnow, 1975*). More recently, a high-resolution model for RNAP-DNA binding energy was determined using data from a massively parallel reporter assay called Sort-Seq (*Kinney et al., 2010*). This position-specific

affinity matrix (PSAM) assumes that the nucleotide at each position contributes additively to the overall binding energy (*Figure 6A*). This model is consistent with previously described RNAP binding motifs but, unlike those motifs, it can predict binding energy in physically meaningful energy units (i. e., kcal/mol). In what follows we denote these binding energies as $\Delta\Delta G_P$, because they describe differences in the Gibbs free energy of binding between two DNA sites.

There is good reason to believe this PSAM to be the most accurate current model of RNAP-DNA binding. However, subsequent work has suggested that the predictions of this model might still have substantial inaccuracies (*Brewster et al., 2012*). To investigate this possibility, we compared our measured values for the Gibbs free energy of RNAP-DNA binding ($\Delta G_P = -k_B T \log P$) to binding energies ($\Delta\Delta G_P$) predicted using the PSAM from *Kinney et al. (2010)*. These values are plotted against one another in *Figure 6B*. Although there is a strong correlation between the predictions of the model and our measurements, deviations of 1 kcal/mol or larger (corresponding to variations in $P$ of 5-fold or greater) are not uncommon. Model predictions also systematically deviate from the diagonal, suggesting inaccuracy in the overall scale of the PSAM.

This finding is sobering: even for one of the best understood DNA-binding proteins in biology, our best sequence-based predictions of in vivo protein-DNA binding affinity are still quite crude. When used in conjunction with thermodynamic models, as in *Kinney et al. (2010)*, the inaccuracies of these models can have major effects on predicted transcription rates. The measurement and modeling of allelic manifolds sidesteps the need to parametrically model such binding energies, enabling the direct inference of Gibbs free energy values for each assayed RNAP binding site.

## Part 3. Strategy: Distinguishing mechanisms of transcriptional activation

*E. coli* TFs can regulate multiple different steps in the transcript initiation pathway (*Lee et al., 2012*; *Browning and Busby, 2016*). For example, instead of stabilizing RNAP binding to DNA, TFs can activate transcription by increasing the rate at which DNA-bound RNAP initiates transcription (*Roy et al., 1998*), a process we refer to as 'acceleration'. CRP, in particular, has previously been

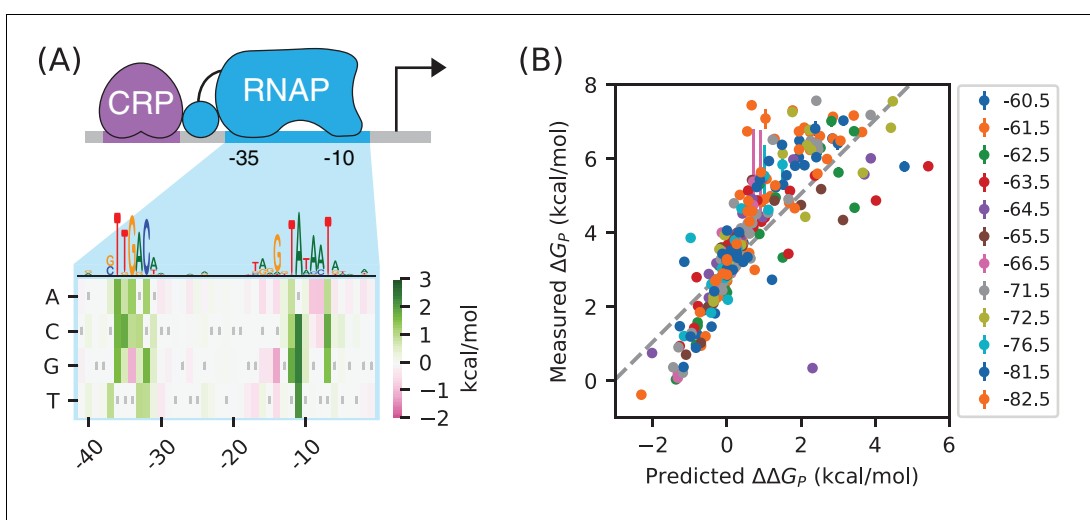

**Figure 6.** RNAP-DNA binding energy cannot be accurately predicted from sequence. (**A**) The PSAM for RNAP-DNA binding inferred by *Kinney et al. (2010)*. This model assumes that the DNA base pair at each position in the RNAP binding site contributes independently to $\Delta G_P$. Shown are the $\Delta\Delta G_P$ values assigned by this model to mutations away from the lac* RNAP site. The sequence of the lac* RNAP site is indicated by gray vertical bars; see also *Appendix 1—figure 1*. A sequence logo representation for this PSAM is provided for reference. (**B**) PSAM predictions plotted against the values $\Delta G_P = -k_B T \log P$ inferred by fitting the allelic manifolds in *Figure 5C*. Error bars on these measurements represent 68% confidence intervals. Note that measured $\Delta G_P$ values are absolute, whereas the $\Delta\Delta G_P$ predictions of the PSAM are relative to the lac* RNAP site, which thus corresponds to $\Delta\Delta G_P = 0$ kcal/mol. The dashed line, provided for reference, has slope 1 and passes through this lac* data point.
DOI: https://doi.org/10.7554/eLife.40618.007

reported to activate transcription in part by acceleration when positioned appropriately with respect to RNAP (*Niu et al., 1996*; *Rhodius et al., 1997*).

We investigated whether allelic manifolds might be used to distinguish activation by acceleration from activation by stabilization. First we generalized the thermodynamic model in *Figure 4A* to accommodate both $\alpha$-fold stabilization and $\beta$-fold acceleration (*Figure 7A*). This is accomplished by using the same set of states and Boltzmann weights as in the model for stabilization, but assigning a transcription rate $\beta t_{\mathrm{sat}}$ (rather than just $t_{\mathrm{sat}}$) to the TF-RNAP-DNA ternary complex. The resulting activated rate of transcription is given by

$$t_+ = t_{\mathrm{sat}} \frac{P}{1+F+P+\alpha FP} + \beta t_{\mathrm{sat}} \frac{\alpha FP}{1+F+P+\alpha FP} + t_{\mathrm{bg}}. \tag{6}$$

This simplifies to

$$t_+ = \beta' t_{\mathrm{sat}} \frac{\alpha' P}{1+\alpha' P} + t_{\mathrm{bg}}, \tag{7}$$

where $\alpha'$ is the same as in *Equation 5* and

$$\beta' = \frac{1+\alpha\beta F}{1+\alpha F} \tag{8}$$

is a renormalized version of the acceleration rate $\beta$. The resulting allelic manifold is illustrated in *Figure 7C*. Like the allelic manifold for stabilization, this manifold has up to five distinct regimes corresponding to different values of $P$ (*Figure 7D*). Unlike the stabilization manifold however, $t_+ \neq t_-$ in the strong RNAP binding regime (regime 5); rather, $t_+ \approx \beta' t_{\mathrm{sat}}$ while $t_- \approx t_{\mathrm{sat}}$.

## Part 3. Demonstration: Mechanisms of class I activation by CRP

We asked whether class I activation by CRP has an acceleration component. Previous in vitro work had suggested that the answer is 'no' (*Malan et al., 1984*; *Busby and Ebright, 1999*), but our allelic manifold approach allows us to address this question in vivo. We proceeded by assaying promoters containing variant alleles of the consensus RNAP binding site (*Figure 8A*). Note that the consensus RNAP site is 1 bp shorter than the lac* RNAP site (*Appendix 1—figure 1*, panel C versus panel B). We therefore positioned the CRP binding site at $-60.5$ bp in order to realize the same spacing between CRP and the $-35$ element of the RNAP binding site that was realized in $-61.5$ bp non-consensus promoters.

The resulting data (*Figure 8B*) are seen to largely fall along the previously measured all-stabilization allelic manifold in *Figure 5B*. In particular, many of these data points lie at the intersection of this manifold with the $t_+ = t_-$ diagonal. We thus find that $\beta \approx 1$ for CRP at $-61.5$ bp. To further quantify possible $\beta$ values, we fit the acceleration model in *Figure 7* to each dataset shown in *Figure 5B*, assuming a fixed value of $t_{\mathrm{sat}} = 15.1$ a.u. The resulting inferred values for $\beta$, shown in *Figure 8C*, indicate little if any deviation from $\beta = 1$. Our high-precision in vivo results therefore substantiate the previous in vitro results of *Malan et al. (1984)* regarding the mechanism of class I activation.

## Part 3. Aside: Surprises in class II regulation by CRP

Many *E. coli* TFs participate in what is referred to as class II activation (*Browning and Busby, 2016*). This type of activation occurs when the TF binds to a site that overlaps the $-35$ element (often completely replacing it) and interacts directly with the main body of RNAP. CRP is known to participate in class II activation at many promoters (*Keseler et al., 2011*; *Salgado et al., 2013*), including the galP1 promoter, where it binds to a site centered at position $-41.5$ bp (*Adhya, 1996*). In vitro studies have shown CRP to activate transcription at $-41.5$ bp relative to the TSS through a combination of stabilization and acceleration (*Niu et al., 1996*; *Rhodius et al., 1997*).

We sought to reproduce this finding in vivo by measuring allelic manifolds. We therefore placed a consensus CRP site at $-41.5$ bp, replacing much of the $-35$ element in the process, and partially mutated the $-10$ element of the RNAP binding site (*Figure 9A*). Surprisingly, we observed that the resulting allelic manifold saturates at the same $t_{\mathrm{sat}}$ value shared by all class I promoters. Thus, CRP appears to activate transcription in vivo solely through stabilization, and not at all through acceleration, when located at $-41.5$ bp relative to the TSS (*Figure 9B*).

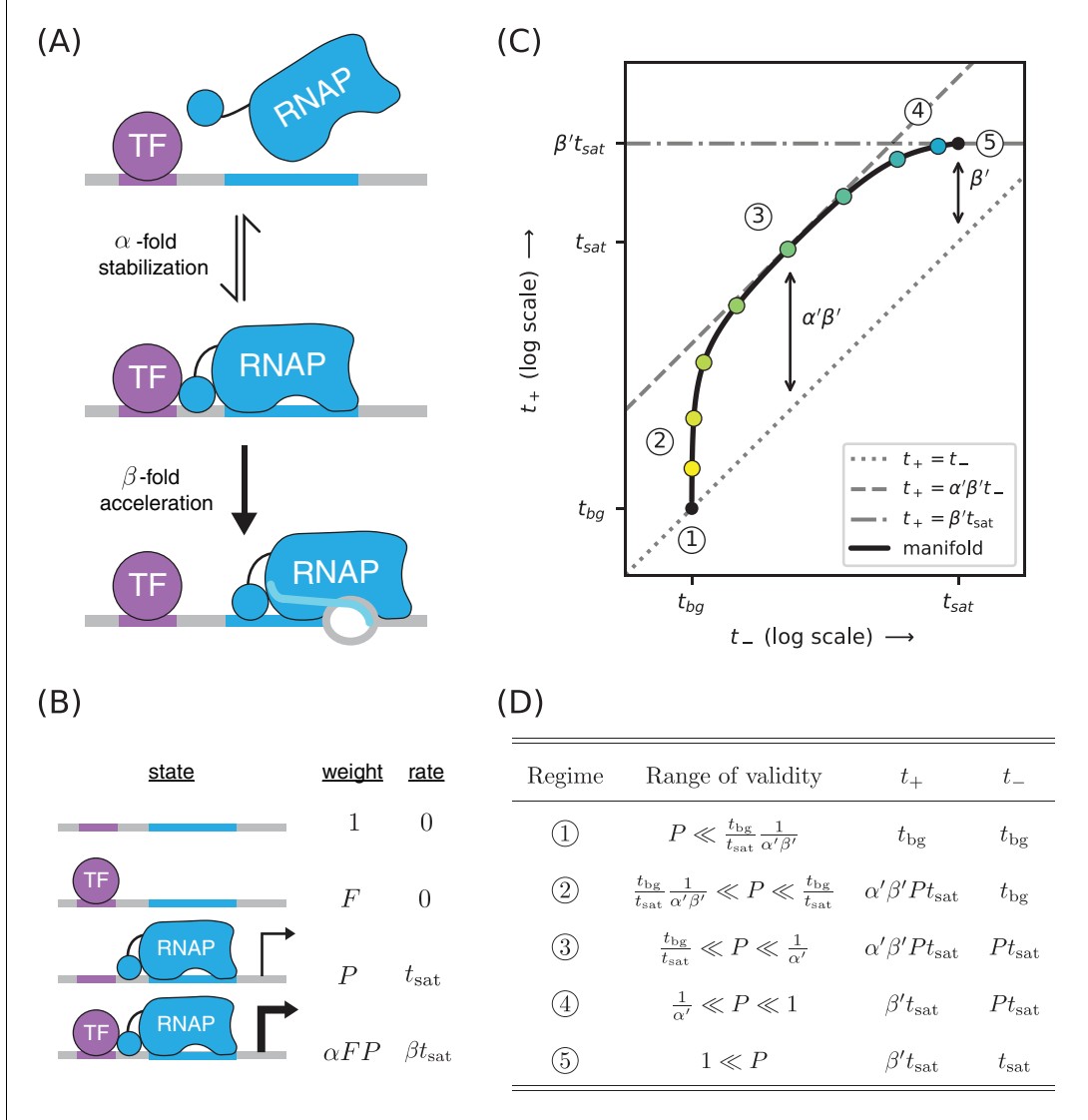

**Figure 7.** A strategy for distinguishing two different mechanisms of transcriptional activation. (**A**) A TF can activate transcription in two ways: by stabilizing the RNAP-DNA complex or by accelerating the rate at which this complex initiates transcripts. (**B**) A thermodynamic model for the dual mechanism of transcriptional activation illustrated in panel A. Note that $\alpha$ multiplies the Boltzmann weight of the doubly bound complex, whereas $\beta$ multiplies the transcript initiation rate of this complex. (**C**) Data points measured as in *Figure 4C* will lie along a 1D allelic manifold having the form shown here. This manifold is computed using $t_+$ values from *Equation 7* and $t_-$ values from *Equation 2*. Note that regime five occurs at a point positioned $\beta'$-fold above the diagonal, where $\beta'$ is related to $\beta$ through *Equation 8*. Measurements in or near the strong promoter regime ($P \gtrsim 1$) can thus be used to determine the value of $\beta'$ and, consequently, the value of $\beta$. (**D**) The five regimes of this allelic manifold are listed.
DOI: https://doi.org/10.7554/eLife.40618.008

The genome-wide distribution of CRP binding sites suggests that CRP also participates in class II activation when centered at −40.5 bp (*Keseler et al., 2011*; *Salgado et al., 2013*). When assaying this promoter architecture, however, we obtained a 2D scatter of points that did not collapse to any discernible 1D allelic manifold (*Figure 9D*). Some of these promoters exhibit activation, some exhibit repression, and some exhibit no regulation by CRP.

These observations complicate the current understanding of class II regulation by CRP. Our in vivo measurements of CRP at −41.5 bp call into question the mechanism of activation previously discerned using in vitro techniques. The scatter observed when CRP is positioned at −40.5 bp suggests

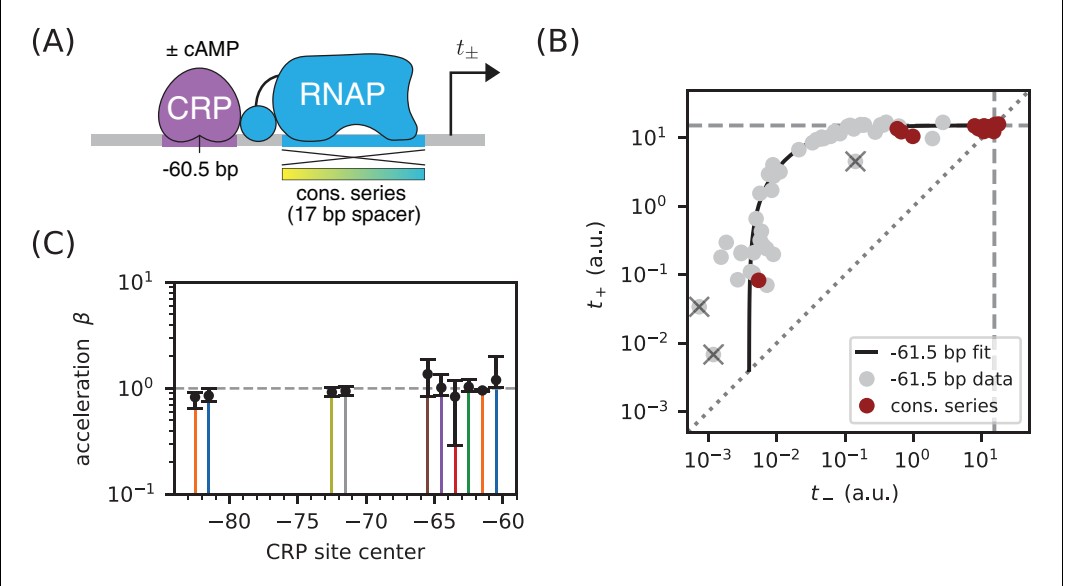

**Figure 8.** Class I activation by CRP occurs exclusively through stabilization. (A) $t_+$ and $t_-$ were measured for promoters containing variants of the consensus RNAP binding site as well as a CRP binding site centered at $-60.5$ bp. Because the consensus RNAP site is 1 bp shorter than the RNAP site of the lac* promoter, CRP at $-60.5$ bp here corresponds to CRP at $-61.5$ bp in *Figure 5*. (B) $n = 18$ data points obtained for the constructs in panel A, overlaid on the measurements from *Figure 5B* (gray). The value $t_{sat} = 15.1$ a.u., inferred for *Figure 5C*, is indicated by dashed lines. (C) Values for $\beta$ inferred using the data in *Figure 5* for the 10 CRP positions that exhibited greater than 2-fold inducibility; $\beta$ values at the two other CRP positions ($-66.5$ bp and $-76.5$ bp) were highly uncertain and are not shown. Error bars indicate 68% confidence intervals.

DOI: https://doi.org/10.7554/eLife.40618.009

that, at this position, the $-10$ region of the RNAP binding site influences the values of at least two relevant biophysical parameters (not just $P$, as our model predicts). A potential explanation for both observations is that, because CRP and RNAP are so intimately positioned at class II promoters, even minor changes in their relative orientation caused by differences between in vivo and in vitro conditions or by changes in RNAP site sequence could have a major effect on CRP-RNAP interactions. Such sensitivity would not be expected to occur in class I activation, due to the flexibility with which the RNAP $\alpha$CTDs are tethered to the core complex of RNAP.

## Discussion

We have shown how the measurement and quantitative modeling of allelic manifolds can be used to dissect cis-regulatory biophysics in living cells. This approach was demonstrated in *E. coli* in the context of transcriptional regulation by two well-characterized TFs: RNAP and CRP. Here we summarize our primary findings. We then address some caveats and limitations of the work reported here. Finally, we elaborate on how future studies might be able to scale up this approach using massively parallel reporter assays (MPRAs), including for studies in eukaryotic systems.

### Summary

In each of our experiments, we quantitatively measured transcription from an allelic series of variant RNAP binding sites, each site embedded in a fixed promoter architecture. Two expression measurements were made for each variant promoter: $t_+$ was measured in the presence of the active form of CRP, while $t_-$ was measured in the absence of active CRP. This yielded a data point, $(t_-, t_+)$, in a two-dimensional measurement space. We had expected the data points thus obtained for each allelic series to collapse to a 1D curve (the allelic manifold), with different positions along this manifold corresponding to different values of RNAP-DNA binding affinity. Such collapse was indeed observed in all but one of the promoter architectures we studied. By fitting the parameters of quantitative

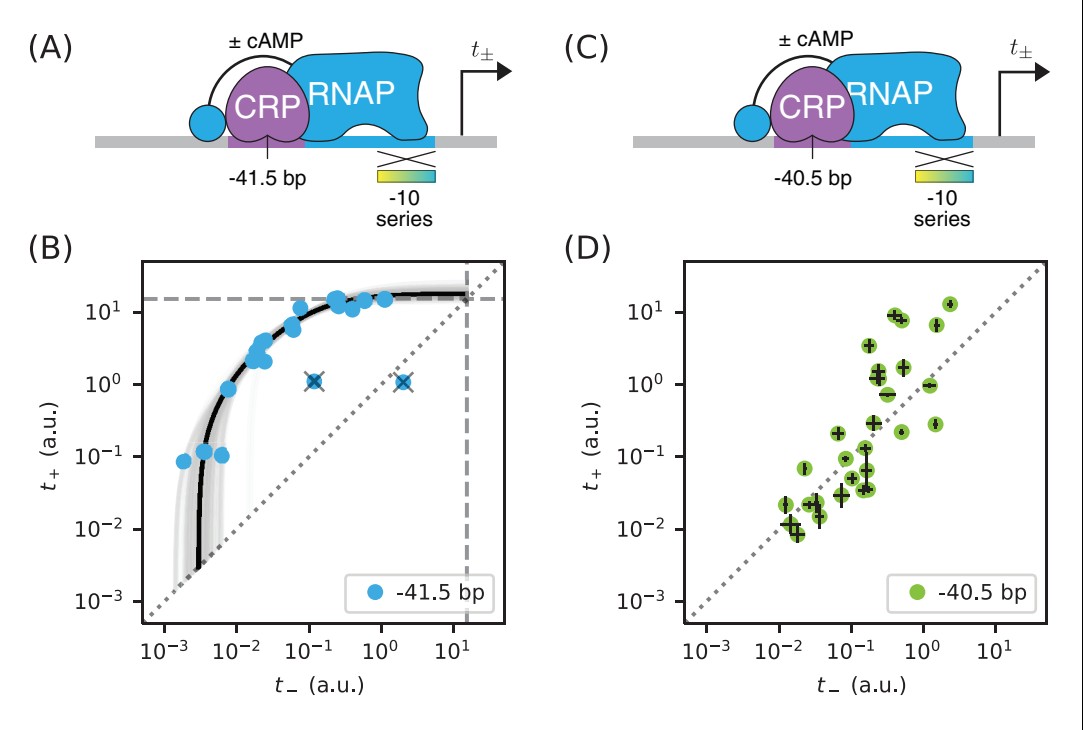

**Figure 9.** Surprises in class II regulation by CRP. (**A**) Regulation by CRP centered at $-41.5$ bp was assayed using an allelic series of RNAP binding sites that have variant $-10$ elements (gradient). (**B**) The observed allelic manifold plateaus at the value of $t_{sat} = 15.1$ a.u. (dashed lines) determined for *Figure 5B*, thus indicating no detectable acceleration by CRP. This lack of acceleration is at odds with prior in vitro studies (*Niu et al., 1996*; *Rhodius et al., 1997*). (**C**) Regulation by CRP centered at $-40.5$ bp was assayed in an analogous manner. (**D**) Unexpectedly, data from the promoters in panel C do not collapse to a 1D allelic manifold. This finding falsifies the biophysical models in *Figures 4A* and *7B* and indicates that CRP can either activate or repress transcription from this position, depending on as-yet-unidentified features of the RNAP binding site. Error bars in panel D indicate 95% confidence intervals estimated from replicate experiments.
DOI: https://doi.org/10.7554/eLife.40618.010

biophysical models to these data, we obtained in vivo values for the Gibbs free energy ($\Delta G$) of a variety of TF-DNA and TF-TF interactions.

In Part 1, we showed how measuring allelic manifolds for promoters in which a DNA-bound TF occludes RNAP can allow one to precisely measure the $\Delta G$ of TF-DNA binding. We demonstrated this strategy on promoters where CRP occludes RNAP, thereby obtaining the $\Delta G$ for a CRP binding site that was used in subsequent experiments. As an aside, we demonstrated how performing such measurements in different concentrations of the small molecule cAMP allowed us to quantitatively measure in vivo changes in active CRP concentration.

In Part 2, we showed how allelic manifolds can be used to measure the $\Delta G$ of TF-RNAP interactions. We used this strategy to measure the stabilizing interactions by which CRP up-regulates transcription at a variety of class I promoter architectures. Our strategy consistently yielded $\Delta G$ values with an estimated precision of $\sim 0.1$ kcal/mol. As an aside, we showed how $\Delta G$ values for RNAP-DNA binding could also be obtained from these data. Notably, these $\Delta G$ measurements for RNAP-DNA binding were seen to deviate substantially from sequence-based predictions using an established position-specific affinity matrix (PSAM) for RNAP. This highlights just how difficult it can be to accurately predict TF-DNA binding affinity from DNA sequence.

In Part 3, we showed how allelic manifolds can allow one to distinguish between two potential mechanisms of transcriptional activation: 'stabilization' (a.k.a. 'recruitment') and 'acceleration'. Applying this approach to the data from Part 2, we confirmed (as expected) that class I activation by CRP does indeed occur through stabilization and not acceleration. As an aside, we pursued this

**Table 1.** Summary of results for class I activation by CRP.

The $\alpha$ and $\Delta G_\alpha$ values listed here correspond to the values plotted in **Figure 5D**. The corresponding value inferred for the saturated transcription rate is $t_{\text{sat}} = 15.1^{+0.6}_{-0.5}$ a.u. Error bars indicate 68% confidence intervals; see Appendix 3 for details. $n$ is the number of data points used to infer these values, while 'outliers' is the number of data points excluded in this analysis. For comparison we show the fold-activation measurements (i.e., $t_+/t_-$) reported in **Gaston et al. (1990)** and **Ushida and Aiba (1990)**; '-' indicates that no measurement was reported for that position.

| Position (bp) | $n$ | Outliers | $\Delta G_\alpha$ (kcal/mol) | $\alpha$ | $t_+/t_-$ (Gaston) | $t_+/t_-$ (Ushida) |
|---|---|---|---|---|---|---|
| −60.5 | 21 | 0 | −2.09 ± 0.08 | $29.6^{+4.7}_{-3.5}$ | 3.85 | - |
| −61.5 | 44 | 3 | −4.10 ± 0.08 | $763^{+113}_{-84}$ | 9.05 | 20.6 |
| −62.5 | 23 | 0 | −2.43 ± 0.11 | $51.4^{+9.0}_{-8.5}$ | 4.22 | - |
| −63.5 | 20 | 1 | −0.88 ± 0.05 | $4.15^{+0.30}_{-0.37}$ | - | - |
| −64.5 | 17 | 0 | −1.08 ± 0.08 | $5.80^{+0.89}_{-0.67}$ | - | - |
| −65.5 | 17 | 0 | −0.48 ± 0.03 | $2.16^{+0.10}_{-0.11}$ | - | - |
| −66.5 | 19 | 1 | 0.00 ± 0.04 | $0.99^{+0.07}_{-0.07}$ | 0.78 | 0.84 |
| −71.5 | 35 | 1 | −2.88 ± 0.04 | $105^{+7}_{-7}$ | 2.50 | 16.4 |
| −72.5 | 20 | 0 | −2.73 ± 0.04 | $83.0^{+5.2}_{-5.8}$ | 3.49 | - |
| −76.5 | 16 | 0 | −0.15 ± 0.04 | $1.27^{+0.09}_{-0.06}$ | 0.54 | - |
| −81.5 | 32 | 0 | −1.53 ± 0.03 | $11.9^{+0.4}_{-0.8}$ | - | - |
| −82.5 | 20 | 0 | −1.82 ± 0.05 | $19.0^{+1.3}_{-1.8}$ | - | 6.99 |

DOI: https://doi.org/10.7554/eLife.40618.011

approach at two class II promoters. In contrast to prior in vitro studies (**Niu et al., 1996**; **Rhodius et al., 1997**), no acceleration was observed when CRP was positioned at −41.5 bp relative to the TSS. Even more unexpectedly, no 1D allelic manifold was observed at all when CRP was positioned at −40.5 bp. This last finding indicates that the variant RNAP binding sites we assayed control at least one functionally important biophysical quantity in addition to RNAP-DNA binding affinity.

## Caveats and limitations

An important caveat is that our $\Delta G$ measurements assume that the *true* transcription rates (of which we obtain only noisy measurements) exactly fall along a 1D allelic manifold of the hypothesized mathematical form. These assumptions are well-motivated by the data collapse that we observed for all except one promoter architecture. But for some promoter architectures, there were a small number of 'outlier' data points that we judged (by eye) to deviate substantially from the inferred allelic manifold. The presence of a few outliers makes sense biologically: the random mutations we introduced into variant RNAP binding sites will, with some nonzero probability, either shift the position of the RNAP site or create a new binding site for some other TF. However, even for promoters that exhibit clear clustering of 2D data around a 1D curve, the deviations of individual non-outlier data points from our inferred allelic manifold were often substantially larger than the experimental noise that we estimated from replicates. It may be that the biological cause of outliers is not qualitatively different from what causes these smaller but still detectable deviations from our assumed model.

The low-throughput experimental approach we pursued here also has important limitations. Each of the 448 variant promoters for which we report data was individually catalogued, sequenced, and assayed for both $t_+$ and $t_-$ in at least three replicate experiments. We opted to use a low-throughput colorimetric assay of $\beta$-galactosidase activity (**Lederberg, 1950**; **Miller, 1972**) because this approach is well established in *E. coli* to produce a quantitative measure of transcription with high precision and high dynamic range. Such assays have also been used by other groups to develop sophisticated biophysical models of transcriptional regulation (**Kuhlman et al., 2007**; **Cui et al., 2013**). However, this low-throughput approach has limited utility because it cannot be readily scaled up.

Our reliance on cAMP as a small molecule effector of CRP presents a second limitation. In our experiments, we controlled the in vivo activity of CRP by growing a specially designed strain of *E. coli* in either the presence (for $t_+$) or absence ($t_-$) of cAMP. This mirrors the strategy used by *Kuhlman et al. (2007)*, and the validity of this approach is attested to by the calibration data shown in *Appendix 2—figure 1*. However, controlling in vivo TF activity using small molecules has many limitations. Most TFs cannot be quantitatively controlled with small molecules, and those that can often require special host strains (e.g., see *Kuhlman et al., 2007*). Moreover, varying the in vivo concentration of a TF can affect cellular physiology in ways that can confound quantitative measurements.

## Outlook

MPRAs performed on array-synthesized promoter libraries should be able to overcome both of these experimental limitations. Current MPRA technology is able to quantitatively measure gene expression for $\geq 10^4$ transcriptional regulatory sequences in parallel. We estimate that this would enable the simultaneous measurement of $\sim 10^2$ highly resolved allelic manifolds, each manifold representing a different promoter architecture. Moreover, by using array-synthesized promoters in conjunction with MPRAs, one can measure $t_+$ and $t_-$ by systematically altering the DNA sequence of TF binding sites, rather than relying on small molecule effectors of each TF. This capability would, among other things, enable biophysical studies of promoters that have multiple binding sites for the same TF; in such cases it might make sense to use measurement spaces having more than two dimensions.

Will allelic manifolds be useful for understanding transcriptional regulation in eukaryotes? Both Sort-Seq MPRAs (*Sharon et al., 2012*; *Weingarten-Gabbay et al., 2017*) and RNA-Seq MPRAs (*Melnikov et al., 2012*; *Kwasnieski et al., 2012*; *Patwardhan et al., 2012*) are well established in eukaryotes so, on a technical level, experiments analogous to those described here should be feasible. The bigger question, we believe, is whether the results of such experiments would be interpretable. Eukaryotic transcriptional regulation is far more complex than transcriptional regulation in bacteria. Still, we believe that pursuing the measurement and modeling of allelic manifolds in this context is worthwhile. Despite the underlying complexities, simple 'effective' biophysical models might work surprisingly well. Similar approaches might also be useful for studying other eukaryotic regulatory processes that are compatible with MPRAs, such as alternative splicing (*Wong et al., 2018*).

Based on these results, we advocate a very different approach to dissecting cis-regulatory grammar than has been pursued by other groups. Rather than attempting to identify a single quantitative model that can explain regulation by many different arrangements of TF binding sites (*Gertz et al., 2009*; *Sharon et al., 2012*; *Mogno et al., 2013*; *Smith et al., 2013*; *Levo and Segal, 2014*; *White et al., 2016*), we suggest focused studies of the biophysical interactions that result from *specific* TF binding site arrangements. The measurement and modeling of allelic manifolds provides a systematic and stereotyped way of doing this. By coupling this approach with MPRAs, it should be possible to perform such studies on hundreds of systematically varied regulatory sequence architectures in parallel. General rules governing cis-regulatory grammar might then be identified empirically. We suspect that this bottom-up strategy to studying cis-regulatory grammar is likely to reveal regulatory mechanisms that would be hard to anticipate in top-down studies.

## Materials and methods

**Key resources table**

| Reagent type (species) or resource | Designation | Source or reference | Identifiers | Additional information |
|---|---|---|---|---|
| Genetic reagent (*E. coli*) | JK10 | this paper | none | genotype: ΔcyaA ΔcpdA ΔlacY ΔlacZ ΔdksA |

*Continued on next page*

*Continued*

| Reagent type (species) or resource | Designation | Source or reference | Identifiers | Additional information |
|---|---|---|---|---|
| Recombinant DNA reagent | pJK47.419 | this paper | none | cloning vector with BsmBI cut sites, *ccdB* cassette, *lacZ* reporter gene, kanamycin resistance, pSC101 origin |
| Recombinant DNA reagent | pJK48 and variants | this paper | none | reporter plasmids cloned from pJK47.419 |
| Chemical compound | cAMP | Sigma-Aldrich | A9501-1G | Adenosine 3',5'-cyclic monophosphate, 1 gram |
| Chemical compound | IPTG | Sigma-Aldrich | I5502-1G | Isopropyl β-D-1- thiogalactopyranoside, 1 gram |
| Chemical compound | ONPG | Sigma-Aldrich | N1127-5G | 2-Nitrophenyl β-D-galactopyranoside, 5 gram |
| Commercial assay or kit | PureLink Genomic DNA Mini Kit | ThermoFisher | K182001 | none |
| Commercial assay or kit | Nextera XT DNA Library Preparation Kit | Illumina | FC-131–1024 | 24 samples |
| Other | RDM | Teknova | M2105 | growth media: MOPS EZ Rich Defined Medium Kit, 5 liter |
| Other | PopCulture Reagent | MilliporeSigma | 71092–4 | 75 milliliters |
| Other | Breathe-Easier film | USA Scientific | 9123–6100 | sterile, 100 per box |
| Other | Epoch 2 Microplate Spectrophotometer | BioTek | EPOCH2C | none |
| Software | analysis scripts | this paper | none | Available at https://github.com/jbkinney/17_inducibility (copy archived at https://github.com/elifesciences-publications/17_inducibility) |

Appendix 1 describes the media, strains, plasmids, and promoters assayed in this work. Appendix 2 describes the colorimetric *β*-galactosidase activity assay, adapted from *Lederberg (1950)* and *Miller (1972)*, that was used to measure expression levels. Appendix 3 provides details about how quantitative models were fit to these measurements, as well as how uncertainties in estimated parameters were computed. *Supplementary file 1* is an Excel spreadsheet containing the DNA sequences of all assayed promoters, all $t_+$ and $t_-$ measurements used in this work, and all of the parameter values fit to these data, both with and without bootstrap resampling.

## Acknowledgments

We thank Stirling Churchman, Barak Cohen, David McCandlish, Bryce Nickels, and Saurabh Sinha for helpful discussions. We also thank Naama Barkai, Ulrich Gerland, Richard Neher, and one anonymous referee for reviewing this manuscript and providing helpful feedback. This work was supported by a CSHL/Northwell Health Alliance grant to JBK and by NIH Cancer Center Support Grant 5P30CA045508.

## Additional information

### Funding

| Funder | Grant reference number | Author |
|---|---|---|
| National Cancer Institute | 5P30CA045508 | Justin B Kinney |

The funders had no role in study design, data collection and interpretation, or the decision to submit the work for publication.

### Author contributions
Talitha L Forcier, Data curation, Software, Formal analysis, Validation, Investigation, Visualization, Methodology, Writing-review and editing; Andalus Ayaz, Data curation, Validation, Investigation, Methodology, Writing-review and editing; Manraj S Gill, Data curation, Validation, Investigation, Methodology; Daniel Jones, Conceptualization, Investigation, Methodology; Rob Phillips, Supervision, Funding acquisition, Writing—review and editing; Justin B Kinney, Conceptualization, Resources, Data curation, Software, Formal analysis, Supervision, Funding acquisition, Validation, Investigation, Visualization, Methodology, Writing—original draft, Project administration, Writing—review and editing

### Author ORCIDs
Rob Phillips http://orcid.org/0000-0003-3082-2809
Justin B Kinney http://orcid.org/0000-0003-1897-3778

### Decision letter and Author response
Decision letter https://doi.org/10.7554/eLife.40618.022
Author response https://doi.org/10.7554/eLife.40618.023

## Additional files

### Supplementary files
• Supplementary file 1. Numerical results plotted in the Figures and listed in *Table 1*. Please refer to the 'overview' sheet within this workbook for a description of each data sheet therein.
DOI: https://doi.org/10.7554/eLife.40618.012

• Transparent reporting form
DOI: https://doi.org/10.7554/eLife.40618.013

### Data availability
All data used to make the Figures is available in Supplementary file 1. The PSAM for RNAP, previously published by Kinney et al. (2010), is also provided in Supplementary file 1 (with permission). Raw data, processed data, and analysis scripts are also available at https://github.com/jbkinney/17_inducibility (copy archived at https://github.com/elifesciences-publications/17_inducibility). No datasets have been deposited in public databases as part of this work.

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

## Appendix 1

DOI: https://doi.org/10.7554/eLife.40618.014

# Media, strains, plasmids, and promoters

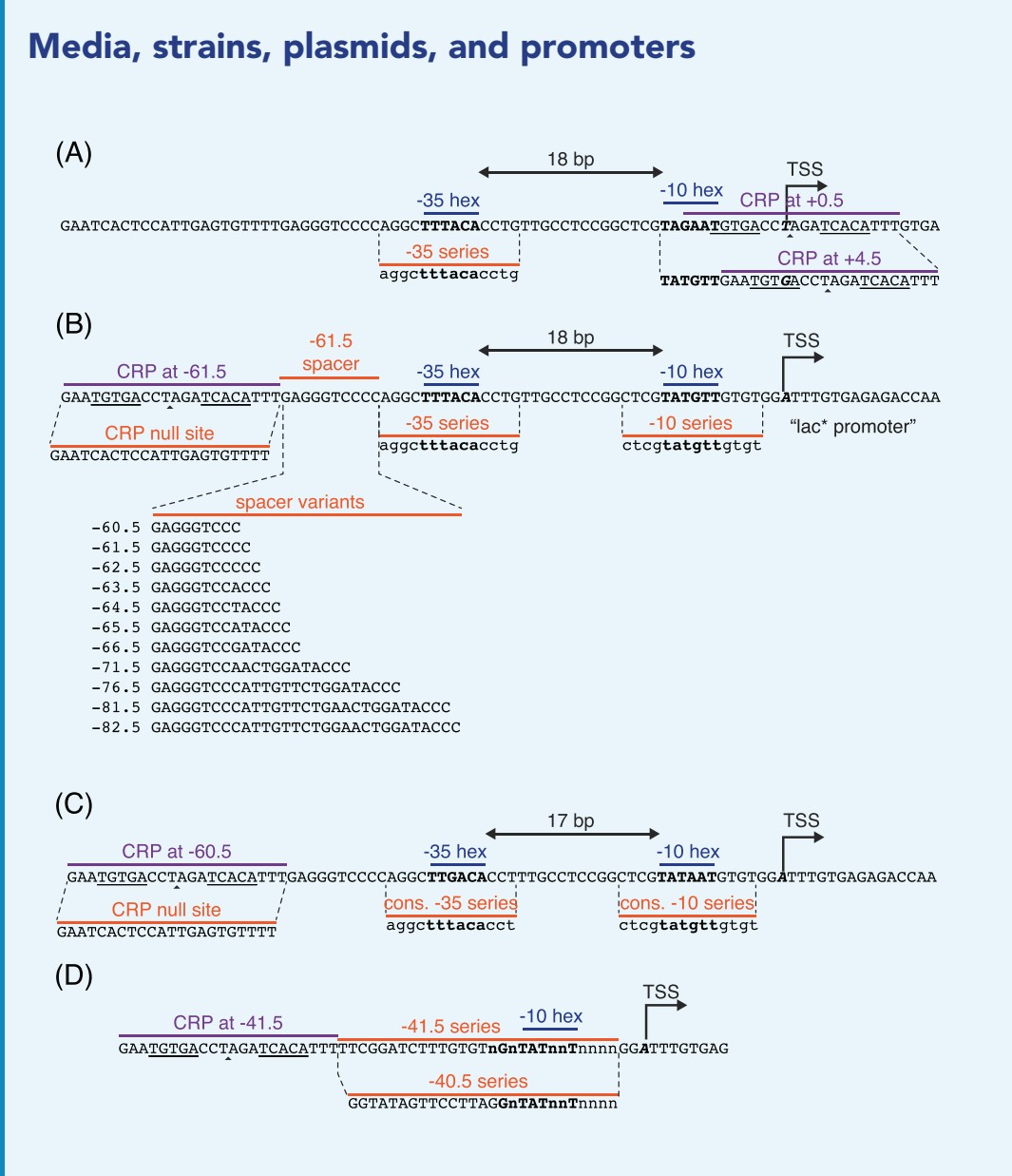

**Appendix 1—figure 1.** Promoter sequences used in this study. In all panels, the −35 and −10 hexamers of the RNAP binding site are in bold. CRP binding site centers are indicated by small triangles. The palindromic pentamers of the core CRP binding site in each construct are underlined. The transcription start site (TSS) is bold and italicized. Lowercase bases ('a','c','g', and 't') indicate positions synthesized with a 24% mutation rate. The lowercase character 'n' indicates completely randomized positions. (**A**) Occlusion promoters assayed for *Figure 2*. (**B**) Class I promoters assayed for *Figure 5*. In the main text we refer to the wild-type promoter with CRP at −61.5 bp as the lac* promoter. The lac* promoter served as the template for all of the promoters shown here. (**C**) Strong class I promoters assayed for *Figure 8*. (**D**) Class II promoters assayed for *Figure 9*.

DOI: https://doi.org/10.7554/eLife.40618.015

Expression measurements were performed on cells grown in rich defined media (RDM; purchased from Teknova) (*Neidhardt et al., 1974*) supplemented with 10 mM NaHCO$_3$, 1 mM IPTG (Sigma), and 0.2% glucose. We refer to this media as RDM'. RDM' was further supplemented with 50 μg/ml kanamycin (Sigma) when growing cells, as well as 250 μM cAMP (Sigma) when measuring $t_+$.

Expression measurements were performed in *E. coli* strain JK10, which has genotype $\Delta cyaA$ $\Delta cpdA$ $\Delta lacZ$ $\Delta lacZ$ $\Delta dksA$. JK10 is derived from strain TK310 (*Kuhlman et al., 2007*), which is $\Delta cyaA$ $\Delta cpdA$ $\Delta lacY$. The $\Delta cyaA$ $\Delta cpdA$ mutations prevent TK310 from synthesizing or degrading cAMP, thus allowing in vivo cAMP concentrations to be quantitatively controlled by adding cAMP to the growth media. Into TK310 we introduced the $\Delta lacZ$ mutation, yielding strain DJ33; this mutation enables the use of $\beta$-galactosidase activity assays for measuring plasmid-based *lacZ* expression. In our initial experiments, we found that the growth rate of DJ33 in RDM' varied strongly with the amount of cAMP added to the media. Fortunately, we isolated a spontaneous knock-out mutation in *dksA* (thus yielding JK10), which caused the growth rate (~ 30 min doubling time) in RDM' to be independent of cAMP concentrations below ~500 μM. We note that JK10 will not grow in minimal media in the absence of cAMP. The TK310, DJ33, and JK10 genotypes were confirmed by whole genome sequencing using the PureLink Genomic DNA Mini Kit (ThermoFisher) for extracting genomic DNA from cultured cells and the Nextera XT DNA Library Preparation Kit (Illumina) for preparing whole-genome sequencing libraries.

Expression of the *lacZ* gene was driven from variants of a plasmid we call pJK48. These reporter constructs were cloned as follows. We started with the vector pJK14 from *Kinney et al. (2010)*. pJK14 contains a pSC101 origin of replication (~ 5 copies per cell; *Thompson et al., 2018*), a kanamycin resistance gene, and a *ccdB* cloning cassette positioned immediately upstream of a *gfpmut2* reporter gene and flanked by outward-facing BsmBI restriction sites. First, the *gfpmut2* gene in this vector was replaced with *lacZ*, yielding pJK47. Next, the ribosome binding site in the 5' UTR of *lacZ* was weakened, yielding pJK47.419; this weakening prevents *lacZ* expression from substantially slowing cell growth in RDM'. pJK47.419 was propagated in DB3.1 *E. coli* (Invitrogen), which is resistant to the CcdB toxin. The promoters we assayed were variants of what we call the 'lac*' promoter. The lac* promoter is similar to the endogenous *lac* promoter of *E. coli* MG1655 except for (i) it contains a CRP binding site with a consensus right pentamer and (ii) it contains mutations that were introduced in an effort to remove previously reported cryptic promoters (*Reznikoff, 1992*). Promoter-containing insertion cassettes were created through overlap-extension PCR and flanked by outward-facing BsaI restriction sites. All primers were ordered from Integrated DNA Technologies. Note that some of the primers used to create these inserts were synthesized using pre-mixed phosphoramidites at specified positions; this is how a 24% mutation rate in the −10 or −35 regions of the RNAP binding site was achieved. The resulting promoter sequences are illustrated in *Appendix 1—figure 1*. To clone variants of pJK48, we separately digested the pJK47.419 vector with BsmBI (NEB) and the appropriate insert with BsaI (NEB). Digests were then cleaned up (Qiagen PCR purification kit) and ligated together in a 1:1 molar ratio for 1 hr using T4 DNA ligase (Invitrogen). After 90 min dialysis, plasmids were transformed into electrocompetent JK10 cells. Individual clones were plated on LB supplemented with kanamycin (50 μg/ml). After initial cloning and plating, each colony was re-streaked, grown in LB+kan, and stored as a catalogued glycerol stock. The promoter region of each clone was sequenced in both directions. Only plasmids with validated promoter sequences were used for the measurements presented in this paper. The promoter sequences of all 448 plasmids used in this study, as well as their measured $t_+$ and $t_-$ values, are provided at https://github.com/jbkinney/17_inducibility (copy archived at https://github.com/elifesciences-publications/17_inducibility).

## Appendix 2

DOI: https://doi.org/10.7554/eLife.40618.014

# Miller assays and the calibration of expression measurements

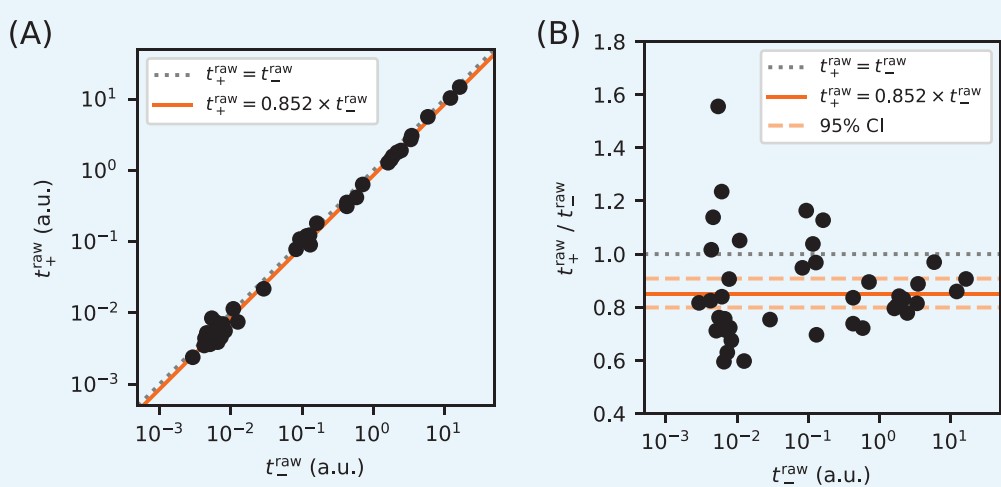

**Appendix 2—figure 1.** Calibration of expression measurements with and without cAMP. (**A**) Measurements of $t_+^{\mathrm{raw}}$ (in 250 µM cAMP) vs $t_-^{\mathrm{raw}}$ (in 0 µM cAMP) for promoters in which the CRP binding site has been replaced by a non-functional 'null' site. As expected, these data lie close to the $t_+^{\mathrm{raw}} = t_-^{\mathrm{raw}}$ diagonal (dotted line). (**B**) Upon closer inspection, however, we found that $t_+^{\mathrm{raw}}$ values consistently fell slightly below corresponding $t_-^{\mathrm{raw}}$ values. Using least-squares fitting we found that, on average, $t_+^{\mathrm{raw}}/t_-^{\mathrm{raw}} = 0.852^{+0.056}_{-0.053}$ where uncertainties indicate a 95% confidence interval (reflecting 1.96 times the standard error of the mean in log space). To correct for this bias, we plot and fit models to $t_+ = t_+^{\mathrm{raw}}$ and $t_- = 0.855 \times t_-^{\mathrm{raw}}$ throughout this paper.

DOI: https://doi.org/10.7554/eLife.40618.017

We obtained $t_+$ and $t_-$ measurements for each promoter as follows. First, the corresponding *E. coli* clone was streaked out on LB+kan agar and grown overnight. A colony was then picked and used to inoculate a 1.5 ml overnight LB+kan liquid culture. Either 8 µl, 6 µl, or 4 µl of the overnight culture were then diluted into 200 µl RDM'+kan. 25 µl of each dilution was then added to 175 µl RDM'+kan in a 96-well optical bottom plate and supplemented with either 0 µM cAMP (for $t_-^{\mathrm{raw}}$), 250 µM cAMP (for $t_+^{\mathrm{raw}}$), or another cAMP concentration (for some $t_+^{\mathrm{raw}}$ measurements in **Figure 3**). The plate was then covered with Breathe-Easier film (USA Scientific) and cells were cultured for ~3 hr at 37 °C, shaking at 900 RPM in a microplate shaker. During this time, 5.5 ml of lysis buffer was freshly prepared using 1.5 ml RDM', 4.0 ml PopCulture reagent (Millipore), 114 µl of 35 mg/ml chloramphenicol (Sigma), and 44 µl of 40 U/µl rLysozyme (Sigma).

Microplate film was removed and cell density (quantified by $A_{600}$) was measured using an Epoch 2 Microplate Spectrophotometer (BioTek). Cells were then lysed by adding 25 µl lysis buffer to each microplate well, incubating the microplate at room temperature for 10 min without shaking, then cooling the microplate at 4 °C for a minimum of 15 min. In each well of a 96-well optical bottom plate, 50 µl of lysate was then added to 50 µl of pre-chilled Z-buffer (**Miller, 1972**) containing 1 mg/ml ONPG (Sigma). Samples were sealed with optical film and both $A_{420}$ and $A_{550}$ were periodically measured in the plate reader over an extended period of time (every 1.5 min for 1 hr or every 15 min for 10 hr, depending on the level of expression expected).

The raw expression levels were quantified from these absorbance data using the formula

$$t_{\pm}^{\text{raw}} = \frac{\Delta A_{420} - \Delta A_{550}}{V \cdot \Delta T \cdot A_{600}}, \tag{9}$$

where $V = 50$ is the volume of lysate in μl added to the ONPG reaction, $\Delta T$ is the change in time from the beginning of the measurement, and $\Delta A_X$ indicates a change in absorbance at $X$ nm over this time interval. Only data from wells with $A_{600} \lesssim 0.5$ were analyzed. Note that the $A_{550}$ term in **Equation 9** is not multiplied by 1.75 as it is in **Miller (1972)**. This is because our $A_{550}$ measurements are used to compensate for condensation on the microplate film, not cellular debris as in **Miller (1972)**; our lysis procedure produces no detectable cellular debris. In practice, **Equation 9** was not evaluated using individual measurements, but was computed from the slope of a line fit to all of the non-saturated absorbance measurements. Raw $A_{420}$, $A_{550}$, and $A_{600}$ values, as well as our analysis scripts, are available at https://github.com/jbkinney/17_inducibility (copy archived at https://github.com/elifesciences-publications/17_inducibility). Median values from at least three independent Miller measurements (and often more) were used to define each measurement shown in the main figures.

Because we controlled the in vivo activity of CRP by supplementing media with or without cAMP, we tested whether CRP-independent promoters produce measurements that vary between these growth conditions. Specifically, we measured $t_{-}^{\text{raw}}$ (in 0 μM cAMP) and $t_{+}^{\text{raw}}$ (in 250 μM cAMP) for 39 promoters in which the CRP binding site was replaced with a 'null' site (see **Appendix 1—figures 1B and C**). These measurements are plotted in **Appendix 2—figure 1**, and show a slight bias. To correct for this bias, we use an unadjusted $t_{+} = t_{+}^{\text{raw}}$ together with an adjusted $t_{-} = 0.855 \times t_{-}^{\text{raw}}$ throughout the main text. Note that $t_{+} = t_{+}^{\text{raw}}$ was used for all nonzero cAMP concentrations, including those in **Figure 3B** that differ from 250 μM. Some upward bias is therefore possible in these $t_{+}$ measurements, but we do not expect this to greatly affect our conclusions.

## Appendix 3

DOI: https://doi.org/10.7554/eLife.40618.014

## Parameter inference

Allelic manifold parameters were fit to measured $t_+$ and $t_-$ values as follows. First, outlier data points were called by eye and excluded from the parameter fitting procedure. We denote the remaining measurements using $t_+^{i,\text{data}}$ and $t_-^{i,\text{data}}$, where $i = 1, 2, \ldots n$ indexes the $n$ non-outlier data points. Corresponding model predictions $t_+^i(\theta)$ and $t_-^i(\theta)$, where $\theta$ denotes model parameters, were then fit to these data using nonlinear least squares optimization. Specifically, we inferred parameters $\theta^* = \text{argmin}_\theta \mathcal{L}(\theta)$ where the loss function is given by

$$\mathcal{L}(\theta) = \sum_{i=1}^{n} \left( \left[ \log \frac{t_+^i(\theta)}{t_+^{i,\text{data}}} \right]^2 + \left[ \log \frac{t_-^i(\theta)}{t_-^{i,\text{data}}} \right]^2 \right). \tag{10}$$

These optimal parameter values $\theta^*$ were used to generate the best-estimate allelic manifolds, which are plotted in black in the main figures. Uncertainties in $\theta$ were estimated by performing the same inference procedure on bootstrap-resampled data. For each variable $X \in \{F, P, \alpha', \beta', t_{\text{sat}}, t_{\text{bg}}\}$, we report

$$X = (X_{50})_{-(X_{50}-X_{16})}^{+(X_{84}-X_{50})} \tag{11}$$

where $X_{50}$, $X_{84}$, and $X_{16}$ respectively denote the median, 84th percentile, and 16th percentile of $X$ values obtained from bootstrap resampling. In the case of $X \in \{F, P, \alpha\}$, we also report

$$\Delta G_X = -k_B T \log X_{50} \pm k_B T \left( \frac{\log X_{84} - \log X_{16}}{2} \right), \tag{12}$$

where 1 kcal/mol = 1.62 $k_B T$ at 37 °C. We now describe each specific inference procedure in more detail.

## Inference for *Figure 2B*

We inferred $\theta = \{t_{\text{sat}}, t_{\text{bg}}, F, P_1, P_2, \ldots, P_n\}$, with model predictions given by

$$t_+^i(\theta) = t_{\text{sat}} \frac{P_i}{1 + F + P_i} + t_{\text{bg}}, \quad t_-^i(\theta) = t_{\text{sat}} \frac{P_i}{1 + P_i} + t_{\text{bg}}. \tag{13}$$

Parameters were fit to the $n = 39$ non-outlier measurements made for promoters with +0.5 bp or +4.5 bp architecture. We found that $F = 23.9_{-2.5}^{+3.1}$ and $t_{\text{bg}} = 2.30 \times 10^{-3}$ a.u., while $t_{\text{sat}}$ values remained highly uncertain.

## Inference for *Figure 3B*

We performed a separate inference procedure for each of the seven cAMP concentrations $C \in \{250, 125, 50, 25, 10, 5, 2.5\}$, indicated in µM units. Specifically, we inferred $\theta_C = \{F_C, P_1, P_2, \ldots, P_{n_C}\}$ where $n_C$ is the number of promoters for which $t_+$ was measured using cAMP concentration $C$. Model predictions were given by

$$t_+^i(\theta_C) = t_{\text{sat}} \frac{P_i}{1 + F_C + P_i} + t_{\text{bg}}, \quad t_-^i(\theta_C) = t_{\text{sat}} \frac{P_i}{1 + P_i} + t_{\text{bg}}, \tag{14}$$

where $t_{\text{sat}} = 15.1$ a.u. is the median saturated transcription rate from *Figure 5C*, and $t_{\text{bg}} = 2.30 \times 10^{-3}$ a.u. is the median background transcription rate from *Figure 2B*. Note that many of the $t_-^i$ measurements were used in the inference procedures for multiple values of $C$, whereas each $t_+^i$ measurement was used in only one such inference procedure.

## Inference for *Figure 5B*

Using data from both the $-10$ and $-35$ allelic series for the $-61.5$ bp promoter architecture, we inferred $\theta = \{t_{\text{sat}}, t_{\text{bg}}, \alpha', P_1, \ldots, P_n\}$. Model predictions were given by

$$t^i_+(\theta) = t_{\text{sat}} \frac{\alpha' P_i}{1 + \alpha' P_i} + t_{\text{bg}}, \quad t^i_-(\theta) = t_{\text{sat}} \frac{P_i}{1 + P_i} + t_{\text{bg}}. \tag{15}$$

For each inferred $\alpha'$, a value for $\alpha$ was computed using $\alpha = \alpha'(1 + F^{-1}) - F^{-1}$, where $F = 23.9$ is the median CRP binding factor inferred for *Figure 2B*.

## Inference for *Figure 5C*

In a single fitting procedure, we inferred $\theta = \{t_{\text{sat}}, t_{\text{bg}}^{-82.5}, \ldots, t_{\text{bg}}^{-60.5}, \alpha'_{-82.5}, \ldots, \alpha'_{-60.5}, P_1, \ldots, P_n\}$ using

$$t^i_+(\theta) = t_{\text{sat}} \frac{\alpha'_{D_i} P_i}{1 + \alpha'_{D_i} P_i} + t_{\text{bg}}^{D_i}, \quad t^i_-(\theta) = t_{\text{sat}} \frac{P_i}{1 + P_i} + t_{\text{bg}}^{D_i}, \tag{16}$$

where each
$D_i \in \{-82.5, -81.5, -76.5, -72.5, -71.5, -66.5, -65.5, -64.5, -63.5, -62.5, -61.5, -60.5\}$
represents the position of the CRP binding site (in bp relative to the TSS) for promoter $i$. Note that a single value for $t_{\text{sat}}$ was inferred for all promoter architectures, while both $t_{\text{bg}}^D$ and $\alpha'_D$ varied with CRP position $D$. The corresponding values of $\alpha$ plotted in *Figure 5D* and listed in the *Table 1* were computed using $\alpha_D = \alpha'_D(1 + F^{-1}) - F^{-1}$ where $F = 23.9$ is the median CRP binding factor inferred for *Figure 2B*. Among other results, we find that $t_{\text{sat}} = 15.1^{+0.6}_{-0.5}$ a.u.

## Inference for *Figure 8C*

For each spacing $D$, we separately inferred $\theta_D = \{\alpha'_D, \beta'_D, t_{\text{bg}}^D\}$ using

$$t^i_+(\theta_D) = \beta'_D t_{\text{sat}} \frac{\alpha'_D P_i}{1 + \alpha'_D P_i} + t_{\text{bg}}^D, \quad t^i_-(\theta_D) = t_{\text{sat}} \frac{P_i}{1 + P_i} + t_{\text{bg}}^D, \tag{17}$$

where $t_{\text{sat}} = 15.1$ a.u. is the median saturated transcription rate inferred for *Figure 5C*. We then computed $\alpha_D = \alpha'_D(1 + F^{-1}) - F^{-1}$ and $\beta_D = \beta'_D(1 + \alpha_D^{-1} F^{-1}) - \alpha_D^{-1} F^{-1}$, using the median CRP binding factor $F = 23.9$ inferred for *Figure 2B*.

## Appendix 4

DOI: https://doi.org/10.7554/eLife.40618.014

# Derivation of allelic manifold regimes

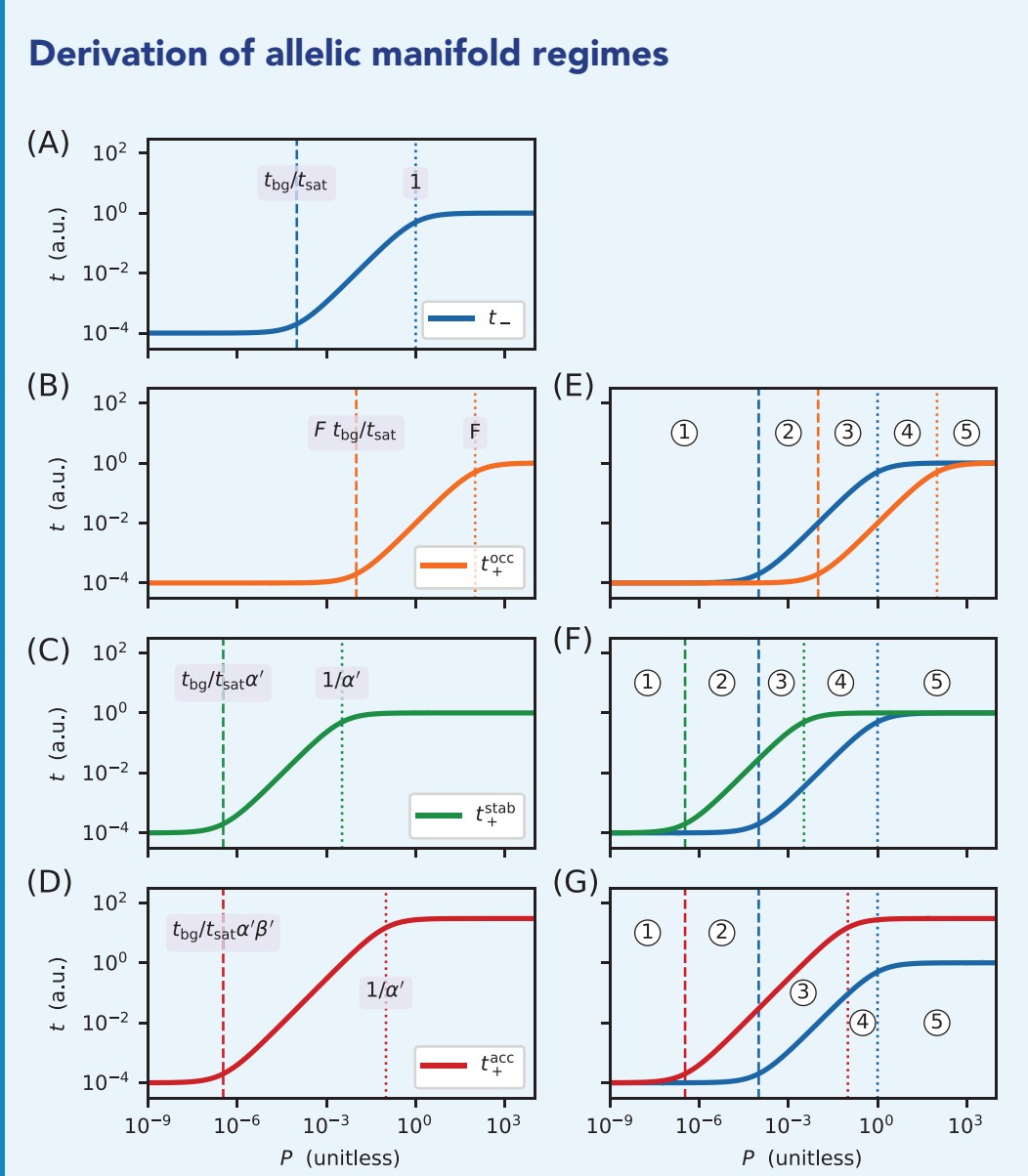

**Appendix 4—figure 1.** Derivation of the regimes of allelic manifolds. Panels A-D show simulated induction curves for transcription $t$ as a function of the RNAP binding factor $P$. Dashed lines indicate boundaries between the minimal and linear regimes of each curve, while dotted lines indicate boundaries between linear and maximal regimes. A formula for the value of $P$ at each regime boundary is also shown. All simulations used $t_{sat} = 1$ a.u., $t_{bg} = 10^{-4}$ a.u., $F = 100$, and $P$ ranging from $10^{-9}$ to $10^4$. (**A**) Induction curve for unregulated transcription; see *Equation 18*. (**B**) Induction curve for transcription repressed by occlusion; see *Equation 19*. (**C**) Induction curve for transcription activated by stabilization ($\alpha = 300$); see *Equation 20*. (**D**) Induction curve for transcription activated by acceleration ($\alpha = 10$, $\beta = 30$); see *Equation 21*. Panels E-G show how overlaps between the six regimes of two induction curves (three for $t_-$ and three for $t_+$) result in five distinct regimes for the corresponding allelic manifold. (**E**) Regimes of the allelic manifold for occlusion, which is shown in *Figure 1C*. (**F**) Regimes of the allelic manifold for stabilization, which is shown in *Figure 4C*. (**G**) Regimes of the allelic manifold for acceleration, which is shown in *Figure 7C*.

DOI: https://doi.org/10.7554/eLife.40618.020

Each transcription rate modeled in this work is a sigmoidal function of the unitless RNAP-DNA binding factor $P$. As such, a log-log plot of transcription $t$ as a function of $P$ reveals a sigmoidal curve having three distinct regimes. The 'minimal' regime of this induction curve comprises values of $P$ that are sufficiently small for $t$ to be well-approximated by its smallest value ($t_{\mathrm{bg}}$ in all cases). The 'maximal' regime occurs when $P$ is so large that $t$ is well-approximated by its largest value (either $t_{\mathrm{sat}}$ or $\beta' t_{\mathrm{sat}}$). Between these maximal and minimal regimes lies a 'linear' regime in which $t$ is approximately proportional to $P$.

For unregulated transcription, which in this paper is denoted $t_-$, these three regimes are given by

$$t_- = t_{\mathrm{sat}}\frac{P}{1+P}+t_{\mathrm{bg}} \approx \begin{cases} t_{\mathrm{bg}} & \text{for } P \ll \frac{t_{\mathrm{bg}}}{t_{\mathrm{sat}}} \\ t_{\mathrm{sat}}P & \text{for } \frac{t_{\mathrm{bg}}}{t_{\mathrm{sat}}} \ll P \ll 1 \\ t_{\mathrm{sat}} & \text{for } 1 \ll P \end{cases} ; \tag{18}$$

see **Appendix 4—figure 1A**. For transcription that is repressed by occlusion (with $F \gg 1$), which we denote here by $t_+^{\mathrm{occ}}$, these three regimes are shifted (relative to $t_-$) to larger values of $P$ by a factor of approximately $F$. As a result,

$$t_+^{\mathrm{occ}} = t_{\mathrm{sat}}\frac{P}{1+F+P}+t_{\mathrm{bg}} \approx \begin{cases} t_{\mathrm{bg}} & \text{for } P \ll F\frac{t_{\mathrm{bg}}}{t_{\mathrm{sat}}} \\ t_{\mathrm{sat}}\frac{P}{1+F} & \text{for } F\frac{t_{\mathrm{bg}}}{t_{\mathrm{sat}}} \ll P \ll F \\ t_{\mathrm{sat}} & \text{for } F \ll P \end{cases} ; \tag{19}$$

see **Appendix 4—figure 1B**. By contrast, for transcription that is activated by stabilization, denoted here by $t_+^{\mathrm{stab}}$, these three regimes shift (relative to $t_-$) to lower values of $P$ by a factor of $1/\alpha'$, giving

$$t_+^{\mathrm{stab}} = t_{\mathrm{sat}}\frac{\alpha'P}{1+\alpha'P}+t_{\mathrm{bg}} \approx \begin{cases} t_{\mathrm{bg}} & \text{for } P \ll \frac{t_{\mathrm{bg}}}{t_{\mathrm{sat}}\alpha'} \\ t_{\mathrm{sat}}\alpha'P & \text{for } \frac{t_{\mathrm{bg}}}{t_{\mathrm{sat}}\alpha'} \ll P \ll \frac{1}{\alpha'} \\ t_{\mathrm{sat}} & \text{for } \frac{1}{\alpha'} \ll P \end{cases} ; \tag{20}$$

see **Appendix 4—figure 1C**. For transcription that is activated partially by acceleration and partially by stabilization, here denoted by $t_+^{\mathrm{acc}}$, two parameters govern the shape of the induction curve. As a result, the boundary between the minimal and linear regimes are shifted (relative to $t_-$) to lower values of $P$ by a factor of $1/\alpha'\beta'$, while the boundary between the linear regime and the maximal regime is shifted down by a factor of only $1/\alpha'$. As a result,

$$t_+^{\mathrm{acc}} = \beta' t_{\mathrm{sat}}\frac{\alpha'P}{1+\alpha'P}+t_{\mathrm{bg}} \approx \begin{cases} t_{\mathrm{bg}} & \text{for } P \ll \frac{t_{\mathrm{bg}}}{t_{\mathrm{sat}}\alpha'\beta'} \\ t_{\mathrm{sat}}\alpha'\beta'P & \text{for } \frac{t_{\mathrm{bg}}}{t_{\mathrm{sat}}\alpha'\beta'} \ll P \ll \frac{1}{\alpha'} \\ t_{\mathrm{sat}}\beta' & \text{for } \frac{1}{\alpha'} \ll P \end{cases} ; \tag{21}$$

see **Appendix 4—figure 1D**.

Each allelic manifold described in the main text has five distinct regimes. These arise from overlaps between the three regimes of $t_-$ and the three regimes of $t_+$. Specifically, the five regimes of the allelic manifold for repression by occlusion, which are listed in **Figure 1D**, arise from the overlaps between the three regimes for $t_-$ and the three regimes for $t_+^{\mathrm{occ}}$. These overlaps are indicated in **Appendix 4—figure 1E**. Similarly, the five regimes of the allelic manifold for activation by stabilization (**Figure 4D**) arise from the overlaps between the regimes of $t_-$ and $t_+^{\mathrm{stab}}$, illustrated in **Appendix 4—figure 1F**, while the regimes of the manifold for activation by acceleration (**Figure 7D**) arise from overlaps between the regimes of $t_-$ and $t_+^{\mathrm{acc}}$, illustrated in **Appendix 4—figure 1G**.

