## [Decision Letter]

[**Editorial note:** This article has been through an editorial process in which the authors decide how to respond to the issues raised during peer review. The Reviewing Editor's assessment is that minor issues remain unresolved.]

Thank you for submitting your article "Precision measurement of cis-regulatory energetics in living cells" for consideration by *eLife*. Your article has been reviewed by three peer reviewers, including Richard A Neher as the Reviewing Editor and Reviewer #1, and the evaluation has been overseen by Naama Barkai as the Senior Editor. The following individual involved in review of your submission has also agreed to reveal their identity: Ulrich Gerland (Reviewer #3). Reviewer #2 remains anonymous.

The Reviewing Editor has highlighted the concerns that require revision and/or responses, and we have included the separate reviews below for your consideration. If you have any questions, please do not hesitate to contact us.

Forcier et al. present a method to quantitatively estimate parameters of transcriptional regulation in vivo. The method is based on a phenomenological model of regulation that accounts for DNA binding of transcription factors and the RNA polymerase as well as interactions between them. The parameters of these models are estimated by comparing transcription for a range of promoter sequences in the presence and absence of the regulator. All reviewers agreed that the authors have devised an innovative and original way to quantify crucial parameters determining the fundamentals of bacterial gene regulation. However, the reviews and the ensuing discussion brought up a number of concerns that the authors should address.

1) How do the inferred parameters depend on growth rates and physiological state of the cells? Given that substantial contributions to the inferred free energies are entropic, changing concentrations of the interacting partners will affect the estimated energies. Comparing parameter inferences at different growth conditions would illuminate the nature of the measured free energies and make the precision of the measurements more interpretable. Repeating the measurements in different growth media would be one way to explore this effect.

2) The outlier classification and removal seem rather arbitrary. While many biophysical aspects change when promoter sequences are exchanged and these factors are difficult to include in a quantitative model, a more thorough discussion of why outliers might arise and how they can be distinguished from data that putatively conforms with the model is necessary.

3) What alternative scenarios might explain the failure of the model when CRP sits at -40.5 (see reviews below). How do you meaningfully distinguish a 'failure to collapse' from 'more outliers'?

Separate reviews (please respond to each point):

*Reviewer #1:*

Forcier et al. present a method to quantitatively estimate parameters of transcriptional regulation in vivo. The method is based on a phenomenological model of regulation that accounts for DNA binding of transcription factors (TF) and the RNA polymerase (RNAP) as well as interactions between TF-RNAP and potential accelerated initiation by TF-RNAP interactions. The parameters of these models are estimated by comparing transcription in the presence and absence of the TF for a variety of the promoter sequences. If a particular scheme of regulation is valid, the two sets of transcription rates are expected to follow a path in the 2d plane and parameters can be estimated from the shape of this path. Failure to collapse indicates a model misspecification.

The nature of the method cancels out/avoids many pitfalls and inaccuracies that arise when fitting more complex explicit models to transcription data. The resulting consistent in vivo measurements are an important step at understanding the energetics of the simplest and most fundamental regulatory systems.

Overall, I feel this is a solid piece of work with consistent results obtained by a clever and original method. The authors discuss ways in which this method could be scaled up to high throughput assays, but this part of the manuscript remains very vague.

The bulk of the DNA-TF binding free energy is claimed to be of entropic nature favoring the unbound state, while RNAP-TF interaction energies are estimated to be much larger than previously thought. To make the manuscript stronger, I would like to see the experiments being performed at different concentrations of the TF (CRP), i.e. vary F via [TF] and not only P.

Minor Comments:

Figure 1C and D: Some of the regimes and their relation to the figure are confusing. All seems correct, but some approximations and definitions are not what I initially thought they were.

a) Why not combine regimes 1 and 2 into t_- = t_bg + Pt_sat and t_+ = t_bg + Pt_sat/(1+F).b) Regime 3: if you don't realize that the figure is meant on a logscale (it is marked, I know, but took me a while to realize). Might be better to mark the diagonal lines as t_-=t_+ and t_- = t_+*(1+F) or similar.

Subsection “Strategy for measuring TF-RNAP interactions in vivo”, second paragraph: cooperatively -> cooperativity

*Reviewer #2:*

The authors carefully quantify binding of the TF crp by changing the affinity of the RNAP binding sites. Experimental measurements of transcription rates are used to infer binding by parameterizing a biophysical model.

The premise of the paper is very interesting and creative. Because of this, I think it is worth investing more energy in making sure the method is made clear – although I am not sure how to do so. The manuscript was also very substantial, and at times difficult to get through. With some clarifications, I think it is worth publishing.

Major comments:

The model and theory.

With the caveat that I do not have strong theoretical expertise:

The use of "manifold" seems to complicate the matter in the context of this paper. As I understand it, the authors are fitting points to a (nonlinear, geometrically complex) model, and looking for deviations from the various regimes of the model. I expect that both t+ and t- are always monotonically increasing functions of P (as they are modelled). A manifold approach would be required if (for example) t+ increased and then decreased as a function of P, but such a case does not appear in the paper, and is not trivial to conceive. Perhaps the authors could explain when/why a manifold approach is necessary.

The classification of points as "outliers" is arbitrary (e.g. Figure 4C for the site at -66.5). A more objective approach could be taken. For example, changes to fitting method could mitigate this. The current loss function minimizes least squares on the log values. I think this is equivalent to assuming error is log normal and minimizing. Could this assumption be relaxed to assume log normal error for most points plus a fraction of points ("outliers") that are drawn from a uniform distribution with some reasonable limits?

Bootstrap: is this necessary? Can the authors infer some confidence interval using the loss function? The most important implication is that the bootstrap procedure may overestimate the precision of their measurements.

Experiments.

I am not sure whether 250uM cAMP is enough to guarantee full occupation of crp in glucose?

Could the authors provide some data for a small number of RNAP binding variants?

It also might be informative to have cAMP induction/repression curves for a few RNAP binding variants.

I'm surprised the authors opted for Miller assays over GFP cytometry (and microscopy). They mention single cell data from another paper, but do not provide any, nor any live cell data. There is information that could be gleaned from single cell data that is relevant. For example, the variance in txn among cells could corroborate the mean txn rate they observe and use to infer binding constants. GFP assays would also provide a corroborative measure of mean expression for the Miller assays.

Can the authors discuss briefly the question of the validity of using plasmid-based assays (in fact I think these are better than chromosomal-based assays for this experiment).

Subsection “Surprises in class II regulation”: The authors frame their result as "When measuring an expression manifold at this position, however, we obtained a scatter of 2D points that did not collapse to any discernible 1D expression manifold (Figure 7D)." I am not convinced. This is a bolder statement than the rest of the paper and requires a bit more evidence to assert. Another interpretation is that for reasons not established, there was more noise (or more outliers) in this experiment than in others.

Minor Comments:

Subsection “Precision measurement of in vivo CRP-DNA binding”: dyadic is an obscure term

*Reviewer #3:*

General assessment:

The authors devise a scheme to systematically determine the effective in vivo interactions between RNA polymerase, transcription factors, and their respective DNA binding sites. The scheme is based on quantitative gene expression measurements from a large number of promoter constructs, which are interpreted using models for competitive and/or cooperative protein-DNA binding. The authors illustrate the power of this approach with a study of cis-regulatory transcription control by the transcription factor CRP in *E. coli*.

The general question attacked by this study is clearly important: How can we characterize the in vivo interactions between DNA-bound proteins that determine transcription rates? The authors make significant progress on this question with their proof-of-principle study of CRP-mediated transcription control, since the underlying approach can readily be transferred and extended to other cases of cis-regulatory transcription control.

Comments and questions to the authors:

1) The authors stress the importance of having in vivo rather than in vitro interaction parameters, and the precision with which they determine these interactions. It is indeed nice to see how well the data collapses, and the quality of the fits is convincing. However, given these encouraging results, I find it important to assess the limitations of both the concept and the precision more broadly. In particular, are the in vivo interaction parameters fixed numbers for a given *E. coli* strain, or do they depend on the state of the cells? All of the experiments were done with the same growth rate and conditions. The effective strength of CRP binding to its consensus DNA site was found to be -2.1 kcal/mol with 0.1 kcal/mol precision under these conditions, but does this parameter change when the cells are put under conditions of e.g. slow growth? The same question applies to the CRP-RNAP interaction. If these parameters do change with the state of the cell, how do the changes compare to the 0.1 kcal/mol precision? This question is crucial to appreciate the significance of the numbers obtained – will they need to be remeasured under every condition or can they be measured just once and then applied to a broad range of conditions?

2) The analysis of CRP regulation from the -40.5 bp site provides an interesting example of a case where the model fails. However, at this point more insight might be gained by considering alternative biophysical models. For instance, could it be that β now depends on the -10 sequence of the promoter? Or could CRP bound in this position generate a situation of "frustrated binding" for RNAP, i.e., it can simultaneously contact the -10 and the -35 region of the promoter when CRP is absent, whereas in the presence of CRP it could only make either the -10 contact or the contact with CRP/-35, and would choose the better one? Perhaps these scenarios are also ruled out by the absence of data collapse – can the authors specify which types of scenarios are ruled out and which are still possible?

3) I think the authors should discuss more clearly which difficulties will need to be overcome when their approach is extended to regulation via more than one TF-binding site. In particular, it seems that determining pairwise interactions may not be enough, since the interaction strength between proteins A and B can depend on whether protein C is bound or not (i.e., 3-body interactions). This can significantly complicate the analysis. How will the approach be generalized – 3-dimensional plots with data collapse onto 2D surfaces? Personally, I think the best hope is that bottom-up approaches like this one will be complemented with top-down approaches like the one described in Hillenbrand et al., *eLife* (2016).

Minor Comments:

– Abstract: in my mind, RNAP is not a TF

– Results section, third paragraph: the conversion from kT to kcal/mol is wrong

– “Our result indicates that, in living cells, this Gibbs free energy is almost entirely canceled by the entropic cost of removing a CRP molecule from the cytoplasmic environment”: can the authors provide a back-of-the envelope estimate to interpret this conclusion – is this approximate cancellation reasonable/expected?

– Second paragraph of subsection “Strategy for measuring TF-RNAP interactions in vivo”: "cooperatively factor" α -> cooperativity α?

---

## [Author Response]

Responses to editorial critiques:

We thank the reviewers and editors for their thoughtful assessment of our manuscript. We have substantially revised this manuscript to address these critiques, as well as to further improve its clarity. Here is a summary of the major changes we have made.

1) We have changed the term “expression manifold” to “allelic manifold”, as we believe our approach is more accurately seen as an extension of the classical genetics concept of allelic series. We have also changed the title of our paper to emphasize the concept of allelic manifolds, and we have added a paragraph to the Introduction aimed at explicitly introducing this concept.

2) We have reduced the length of the Introduction, and we have divided the Discussion into three discrete sections: “Summary”, “Limitations and caveats”, and “Outlook”. We have also organized the Results section into three “Parts”, each of which is sectioned into “Strategy”, “Demonstration”, and “Aside”. We believe these changes will assist the reader in navigating our manuscript.

3) Technical information has been further compartmentalized and expanded in the appendices. These appendices now include a more complete discussion of our model inference methods, a derivation of the five regimes of each allelic manifold, and an added an explanation of how t_+_ and t_-_ measurements were calibrated relative to each other.

Please find below our responses to each specific critique.

1) How do the inferred parameters depend on growth rates and physiological state of the cells? Given that substantial contributions to the inferred free energies are entropic, changing concentrations of the interacting partners will affect the estimated energies. Comparing parameter inferences at different growth conditions would illuminate the nature of the measured free energies and make the precision of the measurements more interpretable. Repeating the measurements in different growth media would be one way to explore this effect.

To address this critique, we assayed occluding promoters using different concentrations of cAMP in the growth media. Our data (in the new Figure 3) suggest that the CRP-DNA binding factor F, and thus the in vivo concentration of active CRP, roughly follows a nontrivial power of cAMP concentration. We note that this nontrivial power-law dependence might result from cooperativity in cAMP-CRP binding, but it might also result from the dynamics of cAMP import and export from cells. Either way, these new data illustrate how allelic manifolds can be used to quantify changes in the in vivo concentrations of TFs.

Our low-throughput experimental setup, however, makes it difficult to repeat our experiments in entirely different growth conditions. Specifically, the *E. coli* strain JK10 required by our experiments will not grow in minimal media in the absence of cAMP, thus precluding measurements in this common growth condition. This illustrates how using a small molecule effector to control TF concentration can be a major limitation. Future work using MPRAs will be able to overcome this hurdle, enabling the measurement of such biophysical parameters in a wide variety of growth conditions. These points are addressed in the revised Discussion.

2) The outlier classification and removal seem rather arbitrary. While many biophysical aspects change when promoter sequences are exchanged and these factors are difficult to include in a quantitative model, a more thorough discussion of why outliers might arise and how they can be distinguished from data that putatively conforms with the model is necessary.

We have clarified the nature of outliers in the revised Discussion. Briefly, because we are introducing random mutations into a promoter sequence, there is a possibility of shifting the RNAP binding site or introducing new binding sites for other TFs. We suspect that this is the primary cause of outliers, which are operationally defined as data points that deviate greatly from the proposed biophysical model.

We decided against implementing a specific mathematical criterion for calling outliers. The suggestion of Referee #2, that we perform Bayesian inference with a stick-and-slab prior, is a reasonable suggestion. But introducing such a model would introduce a substantial complication (in an already lengthy manuscript) while being unlikely to substantively change our results. We note that readers can readily judge whether our outlier designations are reasonable, since all of our data is plotted and shown relative to the fitted manifolds. We expect, however, that an automated method for calling of outliers, like that proposed by Referee #2, will be important in future MPRA-based studies.

3) What alternative scenarios might explain the failure of the model when CRP sits at -40.5 (see reviews below). How do you meaningfully distinguish a 'failure to collapse' from 'more outliers'?

We have clarified our view on this matter in the Discussion. Briefly, in the -40.5 bp architecture, changing the -10 element of the RNAP binding site sequence appears to control a biophysical parameter in addition to RNAP-DNA binding affinity. We suspect that this additional parameter is the strength of the CRP-RNAP interaction; this makes sense structurally, but we do not have additional evidence in support of this hypothesis.

More generally, outliers reflect promoters that substantially deviate from the predictions of the proposed biophysical model. If most promoters in an allelic series are outliers, it means that the proposed biophysical model is of little use and that one should consider an alternative model. There isn’t any fundamental difference between ‘failure to collapse’ and ‘more outliers’. But in all the promoters we investigated, outliers were either very rare or (for CRP at -40.5 bp) were so dominant that no convincing 1D manifold could be visually identified.

Separate reviews (please respond to each point):

Reviewer #1:

*Forcier et al. present a method to quantitatively estimate parameters of transcriptional regulation* in vivo. The method is based on a phenomenological model of regulation that accounts for DNA binding of transcription factors (TF) and the RNA polymerase (RNAP) as well as interactions between TF-RNAP and potential accelerated initiation by TF-RNAP interactions. The parameters of these models are estimated by comparing transcription in the presence and absence of the TF for a variety of the promoter sequences. If a particular scheme of regulation is valid, the two sets of transcription rates are expected to follow a path in the 2d plane and parameters can be estimated from the shape of this path. Failure to collapse indicates a model misspecification.*The nature of the method cancels out/avoids many pitfalls and inaccuracies that arise when fitting more complex explicit models to transcription data. The resulting consistent* in vivo measurements are an important step at understanding the energetics of the simplest and most fundamental regulatory systems.Overall, I feel this is a solid piece of work with consistent results obtained by a clever and original method. The authors discuss ways in which this method could be scaled up to high throughput assays, but this part of the manuscript remains very vague.The revised Discussion section better explains how this assay might be scaled up using MPRAs.The bulk of the DNA-TF binding free energy is claimed to be of entropic nature favoring the unbound state, while RNAP-TF interaction energies are estimated to be much larger than previously thought. To make the manuscript stronger, I would like to see the experiments being performed at different concentrations of the TF (CRP), i.e. vary F via [TF] and not only P.

The new Figure 3 shows expression manifolds measured for multiple [cAMP] concentrations. These data show that F, and thus the active concentration of active CRP in cells, varies as a nontrivial power law of cAMP concentration. We would have liked to pursue additional measurements (e.g., of RNAP-TF interactions) at variable cAMP concentrations, but our paper is already quite lengthy and we do not believe this is essential to support our primary conclusions.

Minor Comments:Figure 1C and D: Some of the regimes and their relation to the figure are confusing. All seems correct, but some approximations and definitions are not what I initially thought they were.a) Why not combine regimes 1 and 2 into t_- = t_bg + Pt_sat and t_+ = t_bg + Pt_sat/(1+F).

Appendix 4 of the revised manuscript provides an explicit derivation of the 5 regimes of each allelic manifold. This should address the referee’s question.

b) Regime 3: if you don't realize that the figure is meant on a logscale (it is marked, I know, but took me a while to realize). Might be better to mark the diagonal lines as t_-=t_+ and t_- = t_+*(1+F) or similar.We have implemented this suggestion in the new Figures 1, 4, and 7.

*Subsection “Strategy for measuring TF-RNAP interactions* in vivo*”, second paragraph: cooperatively -> cooperativity*

Fixed.

Reviewer #2:

The authors carefully quantify binding of the TF crp by changing the affinity of the RNAP binding sites. Experimental measurements of transcription rates are used to infer binding by parameterizing a biophysical model.The premise of the paper is very interesting and creative. Because of this, I think it is worth investing more energy in making sure the method is made clear – although I am not sure how to do so. The manuscript was also very substantial, and at times difficult to get through. With some clarifications, I think it is worth publishing.We have made substantial revisions to this manuscript to improve clarity and readability. In particular, we have renamed “expression manifold” to “allelic manifold”, a concept that is now directly addressed in the Introduction. We have also sectioned the Results and Discussion sections in a way that should help better guide the reader.Major comments:The model and theory.With the caveat that I do not have strong theoretical expertise:The use of "manifold" seems to complicate the matter in the context of this paper. As I understand it, the authors are fitting points to a (nonlinear, geometrically complex) model, and looking for deviations from the various regimes of the model. I expect that both t+ and t- are always monotonically increasing functions of P (as they are modelled). A manifold approach would be required if (for example) t+ increased and then decreased as a function of P, but such a case does not appear in the paper, and is not trivial to conceive. Perhaps the authors could explain when/why a manifold approach is necessary.

We have revised the Introduction to better motivate our adoption of the term “manifold”. The revised Discussion also points out that multi-dimensional generalizations of this concept might be appropriate in situations, e.g., for promoters that contain multiple TF binding sites. It should be noted that expression from some bacterial promoters have been shown to decrease when the RNAP binding site is strengthened due to “trapping” of RNAP by an overly-high affinity binding site, e.g. https://www.ncbi.nlm.nih.gov/pubmed/8006961.

The classification of points as "outliers" is arbitrary (e.g. Figure 4C for the site at -66.5). A more objective approach could be taken. For example, changes to fitting method could mitigate this. The current loss function minimizes least squares on the log values. I think this is equivalent to assuming error is log normal and minimizing. Could this assumption be relaxed to assume log normal error for most points plus a fraction of points ("outliers") that are drawn from a uniform distribution with some reasonable limits?

This is a fair suggestion, but as described above we felt that this would unnecessarily complicate the paper. Such an approach is likely to be required, however, as we transition to high-throughput experiments.

Bootstrap: is this necessary? Can the authors infer some confidence interval using the loss function? The most important implication is that the bootstrap procedure may overestimate the precision of their measurements.

The actual experimental error function is somewhat unclear, and we are reluctant to take our simple Gaussian error assumption too seriously. We used bootstrap resampling primarily because it was the simplest and most transparent thing we could do that would give reasonable results. However, we do expect to use more sophisticated error modeling as we transition to MPRA-based experiments. In the meantime, we have made all of our data (both raw and processed) and analysis code available for researchers who might wish to redo this analysis.

Experiments.I am not sure whether 250uM cAMP is enough to guarantee full occupation of crp in glucose?Could the authors provide some data for a small number of RNAP binding variants?

Our conclusions do not depend on CRP being fully activated by cAMP.

It also might be informative to have cAMP induction/repression curves for a few RNAP binding variants.The new Figure 3 shows how allelic manifolds for occlusion-regulated promoters change in response to variable cAMP concentrations, thus illustrating how changes in the in vivo concentrations of TFs can be quantified.I'm surprised the authors opted for Miller assays over GFP cytometry (and microscopy). They mention single cell data from another paper, but do not provide any, nor any live cell data. There is information that could be gleaned from single cell data that is relevant. For example, the variance in txn among cells could corroborate the mean txn rate they observe and use to infer binding constants. GFP assays would also provide a corroborative measure of mean expression for the Miller assays.

ONPG-based assays of β-galactosidase activity were introduced by Lederberg in 1950 and standardized by Miller in 1972. No other assay of gene expression has a longer track record of providing accurate quantitative measurements. Indeed, as the revised Discussion points out, this is the assay that has been used to establish the most sophisticated biophysical models of transcriptional regulation yet reported.

Can the authors discuss briefly the question of the validity of using plasmid-based assays (in fact I think these are better than chromosomal-based assays for this experiment).

Plasmid-based assays of promoter activity are standard in bacterial transcription field, though on a quantitative level it is unclear how precisely these measurements recapitulate expression from chromosomally-integrated constructs. Unfortunately, our present experimental setup does not allow us to address this question. It is worth emphasizing, though, that being able to systematically dissect cis-regulatory energetics – even just on plasmids – is a substantial advance over present capabilities. Moreover, the proof-of-principle experiments we present here are compatible with genome-integrated MPRAs that have already been developed (https://www.ncbi.nlm.nih.gov/pubmed/29388765), so it should be straightforward to address this question in future work.

Subsection “Surprises in class II regulation”: The authors frame their result as "When measuring an expression manifold at this position, however, we obtained a scatter of 2D points that did not collapse to any discernible 1D expression manifold (Figure 7D)." I am not convinced. This is a bolder statement than the rest of the paper and requires a bit more evidence to assert. Another interpretation is that for reasons not established, there was more noise (or more outliers) in this experiment than in others.

We have added error bars to this plot (now Figure 9D) to indicate the SEM estimated from replicate experiments. These show that the increased scatter is not due to increased measurement noise. The relationship between “no collapse” and “more outliers” is elaborated in the revised Discussion.

Minor Comments:

*Subsection “Precision measurement of* in vivo *CRP-DNA binding”: dyadic is an obscure term*

We have changed “dyadic” to “palindromic” in the revised text.

Reviewer #3:[…] Comments and questions to the authors:

*1) The authors stress the importance of having* in vivo *rather than* in vitro *interaction parameters, and the precision with which they determine these interactions. It is indeed nice to see how well the data collapses, and the quality of the fits is convincing. However, given these encouraging results, I find it important to assess the limitations of both the concept and the precision more broadly. In particular, are the* in vivo *interaction parameters fixed numbers for a given E. coli strain, or do they depend on the state of the cells? All of the experiments were done with the same growth rate and conditions. The effective strength of CRP binding to its consensus DNA site was found to be -2.1 kcal/mol with 0.1 kcal/mol precision under these conditions, but does this parameter change when the cells are put under conditions of e.g. slow growth? The same question applies to the CRP-RNAP interaction. If these parameters do change with the state of the cell, how do the changes compare to the 0.1 kcal/mol precision? This question is crucial to appreciate the significance of the numbers obtained – will they need to be remeasured under every condition or can they be measured just once and then applied to a broad range of conditions?*

These are good questions. Please refer to our response to critique #1 in “Responses to editorial critiques” above.

2) The analysis of CRP regulation from the -40.5 bp site provides an interesting example of a case where the model fails. However, at this point more insight might be gained by considering alternative biophysical models. For instance, could it be that β now depends on the -10 sequence of the promoter? Or could CRP bound in this position generate a situation of "frustrated binding" for RNAP, i.e., it can simultaneously contact the -10 and the -35 region of the promoter when CRP is absent, whereas in the presence of CRP it could only make either the -10 contact or the contact with CRP/-35, and would choose the better one? Perhaps these scenarios are also ruled out by the absence of data collapse – can the authors specify which types of scenarios are ruled out and which are still possible?

This point is clarified in the revised Discussion. Briefly, the lack of collapse suggests that at least two quantitative parameters relevant for expression are changing in response to mutations to the -10 region of the RNAP binding site. We suspect that one of these additional parameters is the CRP-RNAP interaction energy, be we do not have additional evidence of this. DNA-dependence of this interaction could be due to the close proximity between CRP and RNAP when the CRP binds at -40.5 bp. Measurements of this manifold embedded in higher dimensions (e.g., by using 3 cAMP concentrations) might allow us to critically assess this hypothesis, and we plan to pursue this in future work.

3) I think the authors should discuss more clearly which difficulties will need to be overcome when their approach is extended to regulation via more than one TF-binding site. In particular, it seems that determining pairwise interactions may not be enough, since the interaction strength between proteins A and B can depend on whether protein C is bound or not (i.e., 3-body interactions). This can significantly complicate the analysis. How will the approach be generalized – 3-dimensional plots with data collapse onto 2D surfaces? Personally, I think the best hope is that bottom-up approaches like this one will be complemented with top-down approaches like the one described in Hillenbrand et al., eLife (2016).

We believe the best way to address regulation by multiple molecules of a single TF is will be to use MPRAs with array-synthesized oligos. Doing so will allow each individual TF binding site to be turned “on” and “off” independently without any changes to growth conditions. The use of higher-dimensional allelic manifolds might also be useful for this purpose. We expand on this point in the revised discussion.

Minor Comments:– Abstract: in my mind, RNAP is not a TF

Any protein that regulates transcription by binding DNA qualifies as a TF. In fact, in eukaryotes it is quite common to refer to components of the RNA polymerase as TFs (e.g., TFIID).

– Results section, third paragraph: the conversion from kT to kcal/mol is wrong

Thank you for catching this inversion! This was a typo and has been fixed; it does not reflect any errors in our reported results.

– “Our result indicates that, in living cells, this Gibbs free energy is almost entirely canceled by the entropic cost of removing a CRP molecule from the cytoplasmic environment”: can the authors provide a back-of-the envelope estimate to interpret this conclusion – is this approximate cancellation reasonable/expected?

Upon a closer reading of the source reference, we realized that the previously quoted value of -15 kcal/mol does not represent what we thought it had. We have removed this cited quantity. In its place we have added a back-of-the-envelope estimate of what the binding factor F should be based on in vitro measurements of CRP affinity and estimated aqueous CRP concentrations. Our measured F value is far smaller than this estimate. We suggest that this discrepancy is due to the nonspecific binding of CRP and perhaps also to limited DNA accessibility in the cell.

*– Second paragraph of subsection “Strategy for measuring TF-RNAP interactions* in vivo*”: "cooperatively factor" α -> cooperativity α?*

This has been fixed.